# Plastid thylakoid architecture optimizes photosynthesis in diatoms

Serena Flori[1], Pierre-Henri Jouneau[2], Benjamin Bailleul[3], Benoit Gallet[4], Leandro F. Estrozi[4], Christine Moriscot[4], Olivier Bastien[1], Simona Eicke[5], Alexander Schober[6], Carolina Río Bártulos[6], Eric Maréchal[1], Peter G. Kroth[6], Dimitris Petroutsos[1], Samuel Zeeman[5], Cécile Breyton[4], Guy Schoehn[4], Denis Falconet[1] & Giovanni Finazzi[1]

Photosynthesis is a unique process that allows independent colonization of the land by plants and of the oceans by phytoplankton. Although the photosynthesis process is well understood in plants, we are still unlocking the mechanisms evolved by phytoplankton to achieve extremely efficient photosynthesis. Here, we combine biochemical, structural and *in vivo* physiological studies to unravel the structure of the plastid in diatoms, prominent marine eukaryotes. Biochemical and immunolocalization analyses reveal segregation of photosynthetic complexes in the loosely stacked thylakoid membranes typical of diatoms. Separation of photosystems within subdomains minimizes their physical contacts, as required for improved light utilization. Chloroplast 3D reconstruction and *in vivo* spectroscopy show that these subdomains are interconnected, ensuring fast equilibration of electron carriers for efficient optimum photosynthesis. Thus, diatoms and plants have converged towards a similar functional distribution of the photosystems although via different thylakoid architectures, which likely evolved independently in the land and the ocean.

[1] Université Grenoble Alpes (UGA), Laboratoire de Physiologie Cellulaire et Végétale, UMR 5168, Centre National de la Recherche Scientifique (CNRS), Commissariat à l'Energie Atomique et aux Energies Alternatives (CEA), Institut National de la Recherche Agronomique (INRA), Institut de Biosciences et Biotechnologie de Grenoble (BIG), CEA-Grenoble, 38000 Grenoble, France. [2] Laboratoire d'Etudes des Matériaux par Microscopie Avancée, Institut Nanosciences et Cryogénie, Service de Physique des Matériaux et Microstructures, CEA-Grenoble, 38000 Grenoble Cédex 9, France. [3] Institut de Biologie Physico-Chimique (IBPC), UMR 7141, CNRS and Université Pierre et Marie Curie (UPMC), 75005 Paris, France. [4] CNRS, UMR 5075 CNRS, CEA, UGA, Institut de Biologie Structurale, 38000 Grenoble, France. [5] Plant Biochemistry, Department of Biology, ETH Zurich, CH-8092 Zürich, Switzerland. [6] Department of Biology, University of Konstanz, 78457 Konstanz, Germany. Correspondence and requests for materials should be addressed to D.F. (email: denis.falconet@cea.fr) or to G.F. (email: giovanni.finazzi@cea.fr).

Photosynthesis is a unique process that converts sunlight energy into organic matter on Earth, feeding almost the entire food chain. Photosynthesis is accomplished on the land, which is dominated by plants, and in the ocean, which is mostly colonized by phytoplankton. In eukaryotes, this process occurs in a specialized organelle: the plastid. Plant photosynthetic plastids (chloroplasts) are derived from a cyanobacterium-like organism via primary endosymbiosis, whereas the majority of phytoplankton plastids are derived from a red eukaryotic microalga via secondary endosymbiosis. Their different phylogenetic origins have led to distinct structural plastid designs. Differences can be observed at the level of the envelope, the membrane system surrounding the stromal space and of the photosynthetic membrane network, the thylakoids. Primary plastids contain a two-membrane envelope, whereas secondary plastids generally have four envelope membranes[1]. Primary plastids also contain differentiated thylakoid domains that segregate the components of the photosynthetic electron flow chain: the two photosystems (PS), which perform light photochemical conversion and the cytochrome $b_6f$, which catalyses electron exchanges between the two PSs. PSII is mostly located in the appressed grana stacks, PSI is mainly found in the non-appressed stroma lamellae whereas the cytochrome $b_6f$ is more homogeneously distributed[2]. The lateral heterogeneity and the consequent physical confinement of the PSs prevents energy withdrawal from PSII by PSI via the thermodynamically favourable energy transfer (energy spillover)[2]. However, this segregation imposes a need for long-range diffusion of intermediary electron carriers (plastoquinones, plastocyanins or soluble cytochromes) between the two domains. Restricted diffusion within the crowded thylakoid membranes and/or in the narrow luminal space are limiting the maximum rate of photosynthetic electron flow in some conditions[3,4].

No thylakoid subdomains are visible in secondary plastids, where available electron micrographs show loose stacks of mostly three thylakoids (sometimes two or four) with few anastomoses in some cases[5,6]. Moreover, while the membrane distribution of a few complexes (PSI and the light harvesting complex, Fucoxanthin Chlorophyll Protein-FCP)[6] is known, no complete picture of the arrangement of the photosynthetic machinery is available to date. Overall, the mechanisms ensuring optimum light absorption and downstream electron flow are still undetermined in secondary plastids, although the organisms containing these plastids are believe to be responsible for ~20% of the global oxygen production[7]. Here, we combine functional, biochemical, immunolocalization analyses with 3D imaging in the diatom *Phaeodactylum tricornutum*, to reveal a sophisticated thylakoid membrane network that orchestrates photosynthetic light absorption and utilization. We show that segregation of the PSs in specific thylakoid subdomains within a functionally seamless space allows balanced light capture without restraining electron flow for optimal photosynthetic activity.

## Results

### Energy spillover in *P. tricornutum*.
The reported loose thylakoid structure of diatoms should promote random distribution of PSI and PSII, thereby favouring PSII to PSI energy spillover via physical contacts between the complexes[2]. Indeed spillover has been earlier reported upon poisoning PSII (refs 8,9) in red algae, considered to be the ancestors of secondary plastids and, more recently, in dinoflagellates (Symbiondinium)[10], which are derived from secondary endosymbiosis. We tested this hypothesis by measuring changes in PSI activity upon inhibition of PSII in *P. tricornutum*. We reasoned that if PSI and PSII are in physical contact (Fig. 1a), inhibition of PSII photochemistry should increase the utilization of PSII-absorbed light by PSI, thus enhancing PSI activity. Conversely, no change in activity is expected if PSI and PSII are separated and do not share their excitation energy, similar to plants (Fig. 1b).

We found that inhibition of PSII with 3-(3,4-dichlorophenyl)-1,1-dimethylurea (DCMU) plus hydroxylamine (HA, Fig. 1c) did not appreciably accelerate PSI activity in *P. tricornutum* cells (Fig. 1d–f). This was revealed by the lack of significant changes in the oxidation rate of $P_{700}$ (the primary donor to PSI, Fig. 1d) and of its cytochrome electron donors (Fig. 1e, see Methods), that is, of the overall pool of PSI donors (Fig. 1f). Similar results were obtained under different light intensities (Supplementary Fig. 2), indicating that if present[11], spillover is of very limited amplitude in *P. tricornutum*. This finding is in line with earlier reports in other diatoms (*Cyclotella meneghiniana*)[12], where absence of spillover can be deduced based on fluorescence lifetime analysis.

### Segregation of photosynthetic complexes in *P. tricornutum*.
Thus, either (i) lipid or biochemical barriers prevent energy exchange between adjacent PSs or (ii) PSI and PSII are physically segregated in different thylakoid domains. To distinguish between the two possibilities, we immunolocalized the two PSs in cells prepared using the Tokuyasu protocol[13], a method that ensures optimum antibody accessibility while preserving membrane structures (Supplementary Fig. 3). We localized PSI using two different antibodies against a core subunit (PsaA, Fig. 2a) and a more peripheral subunit of the complex (PsaC, Supplementary Fig. 4a). We prefentially found this complex in the external, 'peripheral' stromal-facing thylakoid membranes (Fig. 2d, green sectors), in agreement with earlier results[6]. On the other hand, we mainly located PSII in the 'core' thylakoid membranes (Fig. 2d, violet sectors) using two different antibodies (PsbA, Fig. 2b and PsbC, Supplementary Fig. 4b). We also immunolocalized the cytochrome $b_6f$ complex (using the PetA antibody, Fig. 2c), finding that its distribution was similar to that of PSI.

A statistical analysis of 258 micrographs (Principal Component Analysis, Fig. 2e,f, Supplementary Fig. 4d,e and Supplementary Tables 1 and 2) indicated that the barycentre of the PSI, PSII and cyt $b_6f$ complexes' distribution do not localize in the same thylakoid compartments. This analysis confirmed the preferential 'core' location of PSII (black squares in Fig. 2e,f and Supplementary Fig. 4d,e) and the 'peripheral' location of PSI (red circles in Fig. 2e,f and Supplementary Fig. 4d,e). Conversely, while the cyt $b_6f$ complex is more concentrated in the peripheral membranes (cyan triangles in Fig. 2e,f and Supplementary Fig. 4d,e), its distribution is more homogeneous than that of PSI and PSII. Overall, this non-homogeneous distribution of the photosynthetic complexes is reminiscent of previous results in plants[2] and in green algae[14].

We complemented the immunolocalization analyses with biochemical fractionation. In plant thylakoids[15], PSI, which is located in the stromal-exposed thylakoid lamellae, is more prone to solubilization by detergents than PSII, which is buried in the appressed membranes of the grana. We investigated the detergent accessibility of PSs in chloroplasts isolated from *P. tricornutum* cells by exposing them to increasing concentrations of the mild detergent digitonin and analysed the solubilized supernatant and pellet fractions for the presence of the PSs and cyt $b_6f$ by immunoblotting (Supplementary Fig. 5). As shown in Fig. 2g, solubilization of PSI and cytochrome $b_6f$ requires a lower detergent concentration than for PSII, suggesting that PSI and cyt $b_6f$ are located in the stroma-accessible thylakoids while PSII is in the less accessible membranes of the diatom chloroplasts, in agreement with the immunolocalization results.

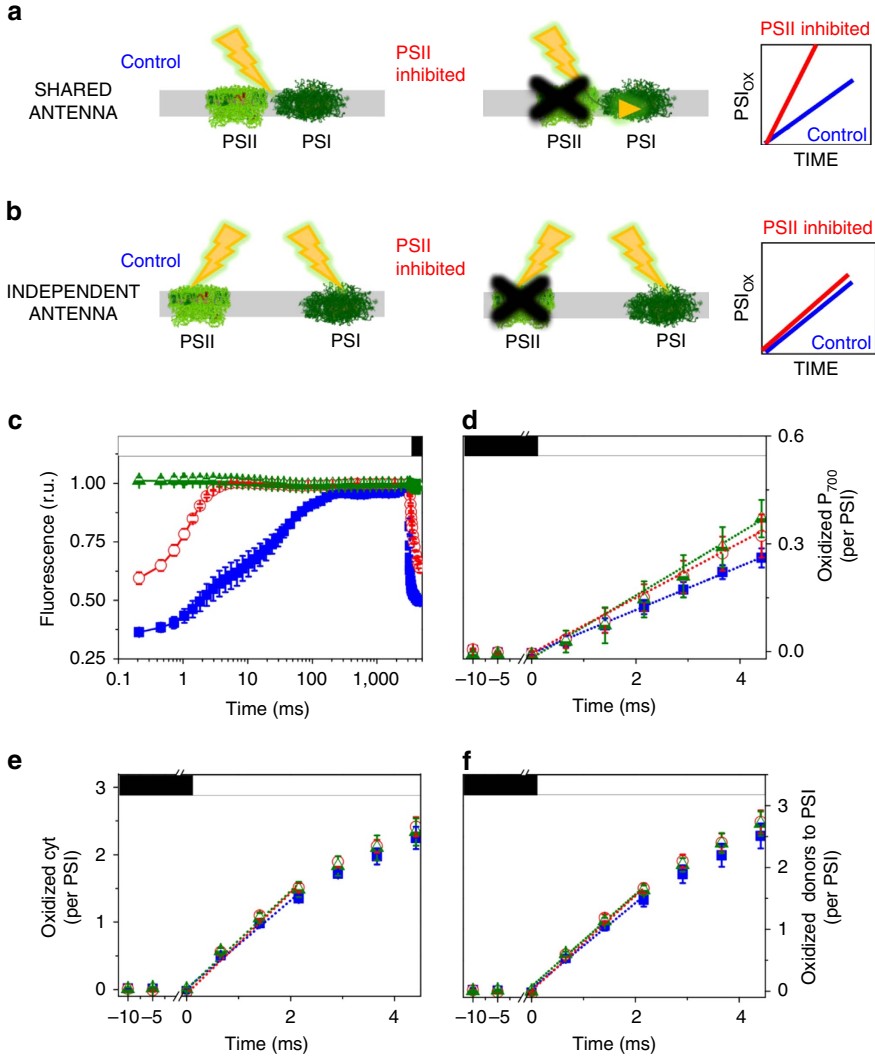

**Figure 1 | Experimental design used to assess energy spillover in diatoms.** Expected effects of energy spillover from PSII to PSI on PSI activity: physical contact between both PSs (**a**), disconnected PSs (**b**). (**c**) Fluorescence emission kinetics confirm full inhibition of PSII by DCMU and HA. (**d**) Kinetics of $P_{700}$ oxidation by light. (**e**) Kinetics of cyt oxidation by light. (**f**) Kinetics of oxidation of the entire pool of PSI electron donors by light. A cyt/PSI ratio of 3 was assumed based on the estimate shown in Supplementary Fig. 1. The light intensity was 800 μmol photons $m^{-2} s^{-1}$. (**c–f**) Solid blue squares: control; empty red circles: 40 μM DCMU; green triangles: 40 μM DCMU + 2 mM HA. Means ± s.e.m. ($n = 6$, from three biological samples). $F_0$: minimum fluorescence emission (active PSII). $F_m$: maximum fluorescence emission (inactive PSII). Closed box: actinic light off. Open box: actinic light on. DCMU and HA were added immediately before measurements.

**Functional consequence of photosynthetic complex segregation.** The segregation of PSI and PSII in different thylakoid sub-compartments should confine the two PSs in slow diffusion domains, as observed in plants[3,4]. We tested this hypothesis using a functional approach[3]. We compared the theoretical ($K_{th}$) and experimental ($K_{exp}$) equilibrium constants between PSI and its electron donors (cytochromes $c_6$ and cytochrome $f$, see Methods). $K_{th}$ was deduced from the redox potentials of cyt $c_6$ (the soluble electron donor to PSI, 349 mV)[16] and $P_{700}$ (the primary electron donor to PSI, 420 mV)[17]. $K_{exp}$ was calculated (equation (2), see Methods) from an 'equilibration plot' (Fig. 3), which shows the relationship between oxidized $P_{700}$ (Fig. 3a) and oxidized c-type cytochromes (cyt, Fig. 3b) during dark re-reduction after illumination (Fig. 3a–c). $K_{exp}$ should be equal to $K_{th}$ in the absence of diffusion domains, but $K_{exp}$ will be less than $K_{th}$ if electron flow is limited by diffusion domains[3,4]. In this second case, the redox state of $P_{700}$ and cyt in each domain will depend on their relative stoichiometry. During the reduction process that follows the light offset, complete reduction of $P_{700}^+$ and a partial reduction of

$cyt^+$ is expected in the compartments with a low $P_{700}$/cyt stoichiometry. Conversely, a large fraction of $P_{700}$ will still be oxidized in domains with a high $P_{700}$/cyt stoichiometry. Because the equilibration plot averages the local redox states of $P_{700}$ and cyt of all the different domains, the concomitant presence of $P_{700}^+$ (in high $P_{700}$/cyt domains) and of reduced cyt c (in low $P_{700}$/cyt domains) translates into a $K_{exp}$ estimate lower than the $K_{th}$ value. We generated several equilibration plots (Fig. 3c) by poisoning photosynthetic electron flow (induced by saturating illumination) with increasing concentrations of DCMU (see also Fig. 3d), and found that diffusion was restricted ($K_{exp} < K_{th}$) when PSII generates more than 150 electrons per second (Fig. 3c, blue and green data points). However, $K_{exp} = K_{th}$ (diffusion is no longer restricted) when the PSII rate is less than 100 electrons per second (Fig. 3c, dark red, red, orange and pink points). Thus, the compartmentalization of PSI and PSII in different thylakoid domains also generates diffusion domains in *P. tricornutum*, similar to plants. However, their equilibration time, 10 ms (corresponding to 100 electrons per second), is much

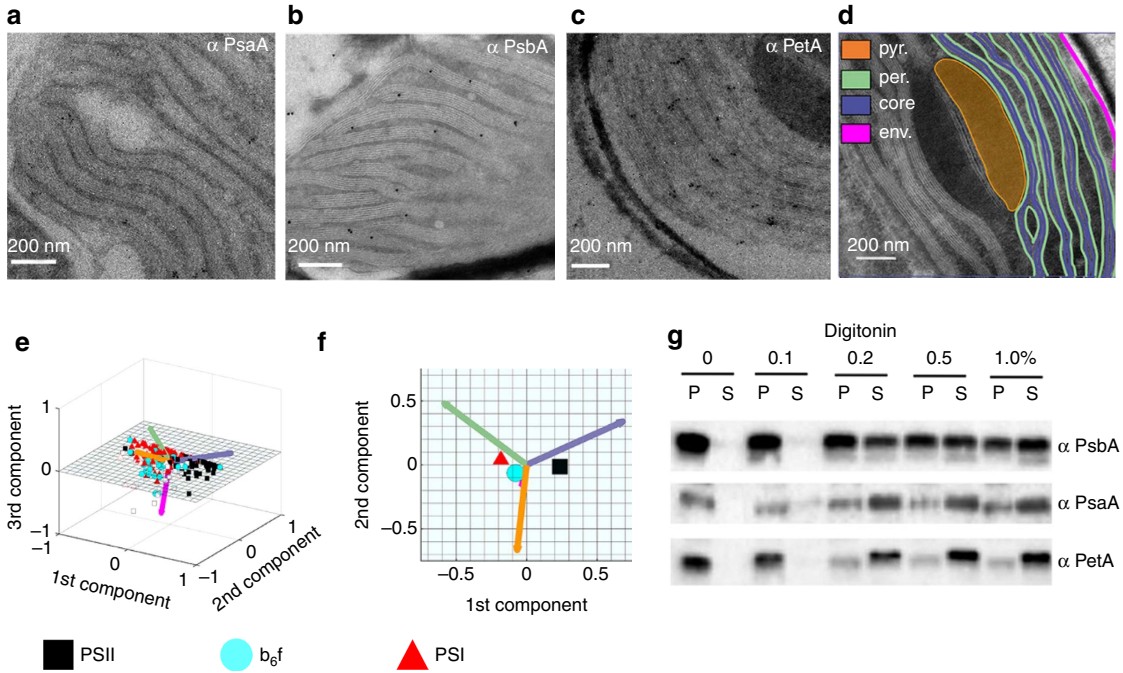

**Figure 2 | Immunolocalization of photosystems and of cyt $b_6f$ in the thylakoid membranes of *P. tricornutum*.** (**a–c**) TEM images of *P. tricornutum* labelled with antibodies directed against the PsaA subunit of PSI (**a**), the PsbA subunit of PSII (**b**) and the PetA subunit of cyt $b_6f$ (**c**). (**d**) TEM micrograph of *P. tricornutum* thylakoid membranes showing four distinct areas: the internal membranes ('core': violet); the external, peripheral membranes ('per.': green); the pyrenoid ('pyr.': orange) and the envelope ('env.': magenta). Bars: 200 nm. (**e**) Principal component analysis of PSI, cyt $b_6f$ and PSII immunolocalization with the PsbA (solid squares), PsbC (open squares), PetA (cyan circle), PsaC (solid triangles) and PsaA (open triangles) antibodies. See also Supplementary Fig. 4. A total of 258 images from four independent cultures were analysed. The first two components represent more than 91% of the variance (see Supplementary Table 1, and Methods for a more detailed explanation). Green arrow: peripheral variable; violet arrow: core variable; orange arrow: pyrenoid variable; Magenta arrow: envelope variable. (**f**) 2D representation of the barycentre for the PSI (α PsaA + α PsaC antibodies, black square), cyt $b_6f$ (PetA, cyan circle) and PSII (α PsbA + α PsbC antibodies, red triangle) distributions. The point size along an axis is proportional to the s.d. along the corresponding component. (**g**) Solubilization of *P. tricornutum* thylakoid membranes with increasing concentrations of digitonin (0.1%, 0.2%, 0.5%, 1%). Pellet (P) and supernatant (S) were analysed by western blotting with the same anti PSI, PSII and cyt $b_6f$ antibodies as in **a–c**. Representative data set of an experiment replicated on three different biological samples.

faster than in plants (~150 ms, corresponding to ~7 electrons per second)[3].

**Chloroplast structure in *P. tricornutum* cells.** To explain the fast equilibration time of redox carriers in diatoms, we re-examined the TEM micrographs of samples prepared with the Tokuyasu protocol. By preserving the membrane structures, this technique allows observing additional features of the *P. tricornutum* thylakoids. We identified regions where thylakoid membranes are apparently interconnected (Supplementary Fig. 6a,b) or where they abruptly 'disappear' in cross-sections (Supplementary Fig. 6c, yellow circles), as if they tilt out of the micrograph plane. These features suggest the existence of a more complex 3D thylakoid network than the simple layout of three loosely juxtaposed thylakoids that is often presented in the case of secondary plastids. We collected 600 images of ultrathin sections using focused ion beam scanning electron microscopy (FIB-SEM) to reconstruct the 3D structure of a *P. tricornutum* cell (Supplementary Movie 1). By segmenting the 3D volume, we identified the organelles and their contacts (Fig. 4a). The mitochondrion (red) appears as a continuous network sitting on the chloroplast (green) with physical contacts between the two organelles (Fig. 4b). This mitochondrial localization likely facilitates energetic exchange between the two organelles, as recently reported[18]. Contact points are also seen between the chloroplast and nucleus (Fig. 4c), as expected since the outer membrane of the chloroplast envelope in secondary plastids is in

connection with the nuclear ER in secondary plastids due to their evolutionary history[19]. These contacts could possibly mediate exchanges between the two compartments, including redox signalling[20] as recently proposed in plants via the formation of transient connections between the chloroplasts and the nucleus, the stromules[21].

The 3D structure of the photosynthetic membranes (Fig. 5a–d) confirmed the presence of parallel layers of stacked thylakoids (purple), but also revealed the presence of connections (Fig. 5b–d, yellow circles) between them. Although the resolution of these images (4 nm pixel, see Methods) does not allow to distinguish the individual thylakoid membranes, we could nonetheless distinguish the connections from plastoglobules, chloroplast lipoprotein particles often observed between the photosynthetic membranes, which appear as globular particles in our 3D reconstruction (Fig. 5c, red circles).

**Discussion**

Our 3D FIB-SEM reconstruction of the *P. tricornutum* plastid thus suggests the existence of an intricate thylakoid network, at variance with previous hypotheses suggesting that the photosynthetic membranes of secondary endosymbiotic plastids are loosely structured[22]. The compartmentalization of the PSs in the peripheral and core thylakoid membranes (Fig. 2) is compatible with the hypothesis that the core membranes are enriched in monogalactosyldiacylglycerol, since this lipid favours the stability and function of the dimeric PSII complex[23,24]. The observed organization of the PSs in the thylakoids accounts for

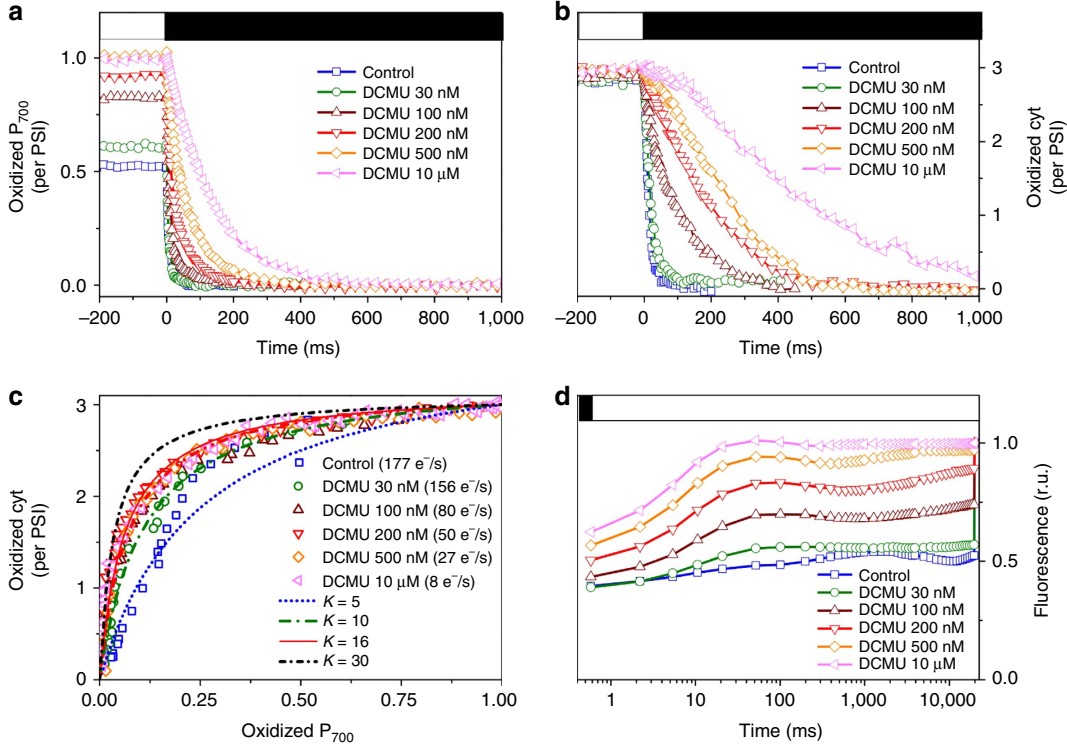

**Figure 3 | Spectroscopic features of the cytochromes and P$_{700}$ components of the electron transfer chain in *P. tricornutum* cells.** (**a–b**) Redox kinetics of P$_{700}$ (**a**) and cyt (**b**) at the offset of a steady state illumination of 800 µmol photons m$^{-2}$ s$^{-1}$ in the absence and in the presence of increasing concentrations of DCMU. Closed box: actinic light off. Open box: actinic light on. (**c**) Equilibrium plots displaying the percentage of oxidized cyt (from **b**) as a function of the percentage of oxidized P$_{700}$ (from **a**). Every point in **c** represents a given time after the offset of light in **a,b**. The dotted lines represent simulations with different values of the equilibrium constant. The rate of electron transfer (calculated as described in Methods) was modified by addition of increasing concentrations of DCMU. (**d**) Fluorescence induction kinetics measured for every DCMU concentration employed in **a,b**. The decrease in variable fluorescence indicates the progressive inhibition of PSII by DCMU. Cells were exposed to 18 µmol photons m$^{-2}$ s$^{-1}$ because no variable fluorescence can be observed at 800 µmol photons m$^{-2}$ s$^{-1}$ even in the absence of DCMU (see, for example, Fig. 1c). DCMU was added immediately before measurements. Blue square: control. Green circle: DCMU 30 nM. Wine upwards triangles: DCMU 100 nM. Red downwards triangles: DCMU 200 nM. Orange losange: DCMU 500 nM. Pink leftwards triangle: DCMU 10 µM. Blue dots: simulation with an equilibrium constant of 5. Green dash and dot line: simulation with an equilibrium constant of 10. Red continuous line: simulation with an equilibrium constant of 16. Black short dash dot line: simulation with an equilibrium constant of 30.

optimum partitioning of absorbed light in low and high light conditions. Limited spillover prevents unbalanced light capture by PSI and PSII, which have similar absorption spectra in diatoms, unlike plants[22]. This may explain why state transitions, the migration of the light harvesting complexes between PSII and PSI to optimize low light capture in plants[25], have been reported to not exist in diatoms[26]. Limited spillover in diatoms could also explain the high capacity of PSII to thermally dissipate excess light through non-photochemical quenching[27]. Indeed, non-photochemical quenching is not expected if the surplus energy in PSII were to be dissipated via spillover to PSI, as in red algae[9].

Our results suggest that *viridiplantae* (including plants and green algae) and diatoms have achieved a similar functional topology of the PSs to optimize photosynthetic light utilization. However, this functional equivalence is achieved with different thylakoid architectures, which likely evolved independently in primary and secondary plastids, and differently affect the electron flow capacity. While PSs confinement constrains electron flow in plants, possibly limiting photosynthesis, no such limitation is observed in diatoms, where the less structured thylakoids allow very fast redox equilibration between the two PSs. Indeed, the presence of connections between thylakoid layers should facilitate diffusion of cyt $c_6$ between the cyt $b_6f$ complexes in the core membranes and the PSI in the peripheral ones, and the diffusion

of plastoquinones from PSII in the core membranes towards the cyt $b_6f$ complexes in the peripheral regions. Overall, the faster diffusion of the soluble electron carriers would promote fast redox equilibration between the photosynthetic complexes in the diatom even at very high electron flux, unlike plants.

We propose that these features, along with the tight interactions between organelles for efficient energetic exchange[18], provide the most adapted framework for high photosynthetic efficiency and acclimation capacity to the ever-changing ocean environment. Indeed, the less 'rigid' structure of secondary plastid could allow the establishment of physical contacts between PSs possible under conditions where substantial protection of PSII is needed. Consistent with this idea, red microalgae can develop sustained spillover to protect PSII in high light[9], while the symbiotic alga Symbiodinium triggers PSII spillover in response to temperature stress[10]. In the latter case, occurrence of topological changes favouring physical contacts between the two PSs has been proposed to account for the enhancement of spillover[10]. On the other hand, accumulation of PSI in specific thylakoids domains has been reported in *P. tricornutum* cells exposed to a particular light regime (prolonged far red light illumination), possibly to segregate it from PSII (ref. 28).

Similar structural features have been reported in green algae. In Chlamydomonas, where the number of stacks can vary from 2 to 15, with a median of 3 thylakoids[29,30], connections between

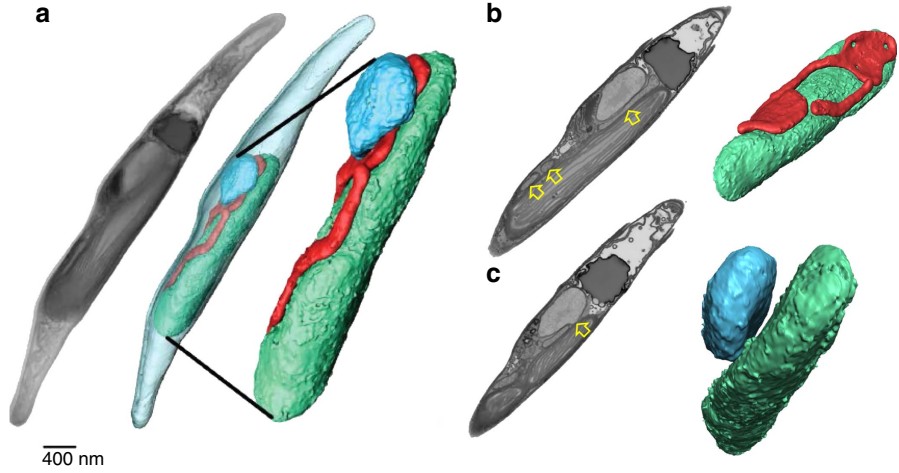

**Figure 4 | Three-dimensional organization of a *P. tricornutum* cell.** (**a**) Whole cell reconstruction of an intact *P. tricornutum* cell based on FIB-SEM images reveals the physical contacts between the chloroplast (green), mitochondrion (red) and nucleus (blue). (**b**) Chloroplast–mitochondria interaction. (**c**) Chloroplast–nucleus interaction. Images represent frames from Supplementary movie 1. Grey pictures in **a**, stacks of SEM micrographs; in **b**,**c**: selected single SEM frame. Coloured pictures in **a**–**c**: 3D reconstruction. Yellow arrows highlight contacts between organelles. Bar: 400 nm.

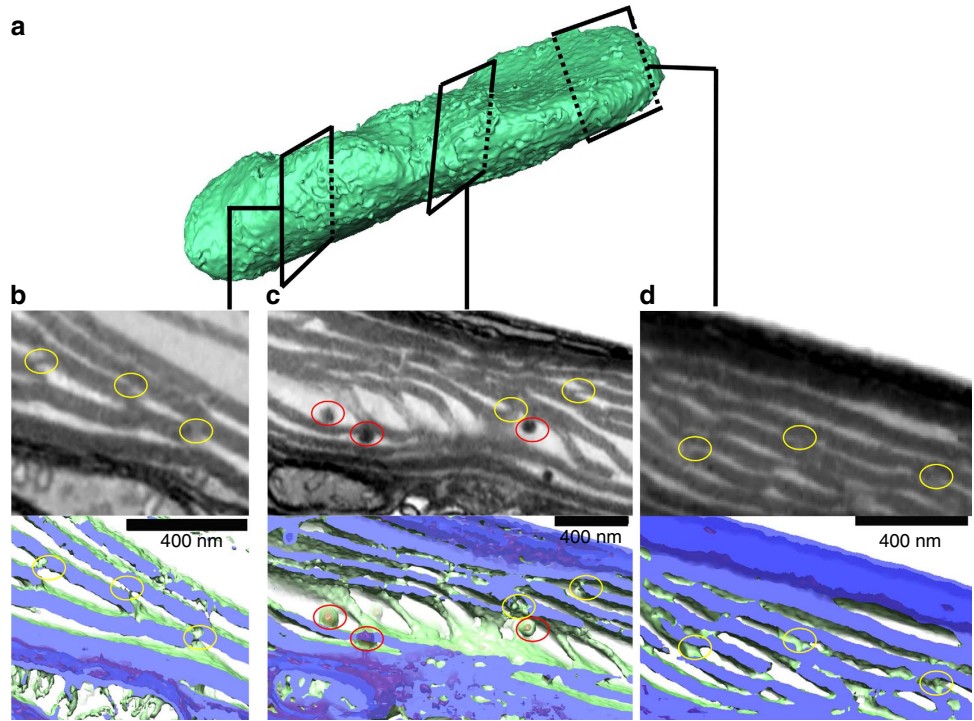

**Figure 5 | Structural arrangement of the photosynthetic membranes in *P. tricornutum*.** (**a**) 3D image of a *P. tricornutum* plastid. Latitudinal (**b**) and longitudinal (**c**,**d**) sections reveal the 3D arrangement between the parallel photosynthetic membranes. Connections between the thylakoid layers (yellow circles) can be clearly differentiated from the plastoglobules (red circles), which appear as globular structures. (**b**–**d**): Top: representative slices of the 3D reconstruction represented as grey-levels (darker is denser). (**b**–**d**) Bottom: same areas as in top panels represented as two isosurfaces. The low density isosurface (green, thylakoid volume) and the high density isosurface (red, plastoglobules) are sliced by a semi-transparent plane (violet), to show the thylakoid stacks. Bars: 400 nm.

thylakoids also appear in cryo-tomograms[29]. While no spillover exists between PSII and PSI in this alga[31], recent data have shown that exposure to different light qualities induces major structural changes in the thylakoids (as revealed by SANS, Small Angle Neutron Scattering), triggering changes in the harvesting capacity of PSII (ref. 32).

## Methods

**Phaeodactylum tricornutum cultivation.** The *P. tricornutum* Pt1 strain (CCAP 1055/3) was obtained from the Culture Collection of Algae and Protozoa, Scottish Marine institute, UK. Cells were grown in the ESAW (Enriched Seawater Artificial Water) medium[33], in 50 ml flasks in a growth cabinet (Certomat BS-1, Sartorius Stedim, Germany), at 19 °C, a light intensity of 20 $\mu$mol photon m$^{-2}$ s$^{-1}$, a 12-h light/12-h dark photoperiod and shaking at 100 r.p.m. Cells were collected in exponential phase, concentrated to a density of $2 \times 10^7$ cells per ml and used for experimental characterization.

**Spectroscopic measurements.** Spectroscopic analysis was performed on intact cells at 20 °C, using a JTS-10 spectrophotometer (Biologic, France). To assess energy spillover from PSII to PSI, redox changes of $P_{700}$ and of its electron donor pool were monitored. Because of the high equilibrium constant between $P_{700}$ and its electron donor pool, one needs to estimate the redox states of both $P_{700}$ and of this pool to quantify the whole amount of electrons that is delivered to PSI. In diatoms, a c-type cytochrome acts as the electron donor to PSI, equivalent to plastocyanin in plants. This cytochrome will be referred as to cytochrome $c_6$ (ref. 34), instead of cytochrome cx (ref. 35) or cytochrome $c_6$A (ref. 36) as used in other publications. Cyt $c_6$ and cytochrome *f* of the cyt $b_6f$ complex have very similar absorption features. It is thus not possible to distinguish them spectroscopically. We define therefore as 'cyt', the pool of cyt $c_6$ + cyt *f*. Cyt redox changes were calculated as $[554] - 0.4 \times [520] - 0.4 \times [566]$, where [554], [520] and [566] are the absorption difference signals at 554, 520 and 566 nm, respectively[18]. $P_{700}$ redox changes were measured at 705 nm. To rule out any possible contribution of fluorescence emission at this wavelength, experiments were repeated at 820 nm (where $P_{700}^+$ is still detected but chlorophyll fluorescence is not measured). Similar results were obtained at both wavelengths, indicating that the interference between $P_{700}$ redox changes and chlorophyll fluorescence emission was negligible.

Kinetics of oxidation of $P_{700}$, cyt of the total donors to PSI oxidation result from concomitant electron injection by PSII and withdrawal by PSI. Inhibiting PSII activity with DCMU also modifies the rate of electron injection into cyt$^+$ and $P_{700}^+$. This translates into an increase of the net oxidation rate of $P_{700}$ and of cyt, which could be misinterpreted as an increase of the PSI activity. Therefore, to calculate the true PSI oxidation rates, we evaluate the reduction rate of this electron donor pool as the slope ($S_D$) of signal relaxation upon switching the light off (Supplementary Fig. 1b). This rate was added to the net oxidation rate, which we estimate from the slope in the light ($S_L$). The sum ($S_L + S_D$) provides the absolute oxidation rate (see Supplementary Fig. 1 for an example in the case of the total PSI electron donor pool).

Inhibition of PSII activity by DCMU and HA was probed following changes in chlorophyll emission from $F_0$ (minimum fluorescence level in which $Q_A$, the primary quinone acceptor of PSII, is oxidized) to the $F_m$ level, in which $Q_A$ is fully reduced because PSII is blocked. As shown in Fig. 1c and Supplementary Fig. 2a–c, light reduces $Q_A$ in DCMU poisoned samples but this inhibitor alone is not sufficient to fully reduce $Q_A$ in the short time (4 ms) employed in our tests to measure oxidation of $P_{700}$ and of cyt. This is particularly evident in low light (for example, Supplementary Fig. 2a, red circles), because at low photon flux the rate of $Q_A$ reduction is diminished. Since full reduction of $Q_A$ is needed to induce spillover, DCMU alone could not be sufficient to probe the occurrence of energy spillover in our experimental conditions. On the other hand, a complete reduction of $Q_A$ is observed in the presence of HA, because this inhibitor prevents re-oxidation of reduced $Q_A$ in PSII (ref. 37). By ensuring $Q_A$ reduction ($F_m$ level) at the beginning of illumination (Fig. 1c, Supplementary Fig. 2a–c green triangles), HA and DCMU ensure optimum conditions to test the occurrence of spillover.

To assess the existence of restricted diffusion domains, we compared the theoretical equilibrium constant ($K_{th}$) between PSI and its electron donors (cyt) with the experimental one ($K_{exp}$), following previous approaches in plants[3] and bacteria[38]. In *P. tricornutum* we calculate a value of 16 for $K_{th}$, based on the redox potentials of cyt $c_6$ (349 mV)[16] and of $P_{700}$ (420 mV)[17]. To evaluate $K_{exp}$ the following equation was used to relate redox changes of $P_{700}$ and of cyt in an 'equilibrium plot' (Fig. 3c).

$$K_{exp} = \frac{[cyt^+] \times [P_{700}]}{[cyt] \times [P_{700}^+]} \quad [1]$$

where $[cyt]$, $[cyt^+]$, $[P_{700}]$ and $[P_{700}^+]$ represent the concentration of the oxidized and reduced form of the cyt and of $P_{700}$ pools. From equation (1) the relationship between the relative amount of oxidized $P_{700}$ and of cyt can be derived as:

$$y = \frac{K_{exp} \times x}{1 + x \times (K_{exp} - 1)}, \quad [2]$$

with

$$y = \frac{cyt^+}{cyt + cyt^+} \quad [3]$$

and

$$x = \frac{P_{700}^+}{P_{700}^+ + P_{700}} \quad [4]$$

Experiments were performed under light saturated conditions, in which photosynthesis is limited by electron flow itself rather than by other factors (that is, light harvesting by PSI and PSII). In these conditions, it is possible to quantify the rate of equilibration between the diffusion domains from measurements of photosynthetic electron flow.

Finally, the P700/cyt stoichiometry was calculated in intact cells exposed to a saturating single turnover laser flash. This flash generates 1 turnover per PSI, leading to oxidation of 1 cyt per PSI. The amount of oxidized cytochrome was quantified 300 $\mu$s after the flash (that is, when P700 is fully re-oxidized by the cytochromes), and compared to the amount of cyt oxidized in continuous light in the presence of DCMU (40 $\mu$M). Because the flash (which generates one positive charge per PSI) oxidizes 33% of the total oxidable cyt pool (per PSI), we conclude that the c-type cytochromes (cyt $c_6$ + cyt *f*)/PSI ratio is $\sim$3.

**Chloroplast purification.** An original protocol was developed to purify intact chloroplasts from *P. tricornutum*. Cells were collected by centrifugation at 5,000*g*, 10 min, 4 °C. The pellet was resuspended gently in 10 ml of isolation buffer (0.5 M Sorbitol; 50 mM Hepes-KOH; 6 mM EDTA; 5 mM MgCl$_2$; 10 mM KCl; 1 mM MnCl$_2$; 1% (w/v) Poly Vinyl Pyrrolidone 40 [K30]; 0.5% BSA; 0.1% cysteine, pH 7.2–7.5) and passed slowly through a French Press at 90 MPa. Ten millilitres of the isolation buffer were added to the mixture of broken cells on ice in the dark before centrifugation at 300*g* for 8 min to remove intact cells and cell debris. The supernatant was collected and subjected to centrifugation at 2,000*g* for 10 min at 4 °C. The pellet containing the chloroplasts was gently resuspended with a soft paint-brush in 2 ml of washing buffer (0.5 M Sorbitol; 30 mM Hepes-KOH; 6 mM EDTA; 5 mM MgCl$_2$; 10 mM KCl; 1 mM MnCl$_2$; 1% PVP 40 [K30]; 0.1% BSA, pH 7.2–7.5) and loaded on a discontinuous Percoll gradient (10, 20, 30%) in the same buffer. After centrifugation (SW41Ti rotor) at 10,000*g* for 35 min, the chloroplast fraction was recovered in the 20% Percoll layer of the gradient, diluted in the washing buffer (without BSA) and subjected to centrifugation at 14,000*g* for 10 min at 4 °C. Chloroplasts were resuspended in washing buffer and intactness was tested with a Clark electrode (Hansatek, UK) using sodium ferricyanide (1.5 mM) as an electron acceptor. Oxygen evolution in saturating light was measured before and after an osmotic shock (induced by incubation for 5 min in the washing buffer without sorbitol). The ratio between the two rates was used to evaluate intactness, which was approximately 70% in our case.

**Membrane solubilization and immunoblot analysis.** To differentially solubilize the two thylakoids compartments (core and peripheral), chloroplasts were incubated at a final chlorophyll concentration of 0.2 mg ml$^{-1}$ for 10 min at 4 °C with digitonin (C$_{56}$H$_{92}$O$_{29}$, Sigma Aldrich) at increasing final concentrations (0.1, 0.2, 0.5 and 1%). Samples were subjected to centrifugation at 100,000*g* for 5 min (rotor TLA-100), supernatants were collected and pellets were resuspended in the same volume of washing buffer without sorbitol. Samples (1.4 $\mu$g chlorophyll) were loaded onto 4–20% polyacrylamide SDS gels and blotted onto nitrocellulose membranes. Antisera against PSI (PsaA and PsaC, subunits of photosystem I, Agrisera, Se, catalogue numbers: AS06172 and AS10939, respectively), PSII (PsbA and PsbC core subunits of PSII, Agrisera, Se, catalogue numbers: AS05084 and AS111787, respectively) and cytochrome $b_6f$ (PetA, Agrisera, Se, catalogue number: AS06119) were detected by ECL using a CCD (charge-coupled device) imager (Chemidock MP Imaging, Bio-Rad, USA). Antibodies were used at a dilution of 1/10,000 (PsaA, PsaC, PsbA and PsbC) or 1/2,000 (PetA) (Supplementary Fig. 5).

**Sample preparation for immunolocalization.** Cells of *P. tricornutum* were fixed in a double-strength fixative (4% (w/v) formaldehyde, 0.4% (v/v) glutaraldehyde) in PHEM buffer (PIPES 60 mM, HEPES 25 mM, EGTA 10 mM, MgCl$_2$ 2 mM; pH 7.0) in an equal volume to the culture medium (ESAW), and then diluted into a standard strength fixative (2% (w/v) formaldehyde (EMS, USA) and 0.2% (v/v) glutaraldehyde (EMS, USA). After 15 min, fresh standard strength fixative was replaced and fixation proceeded for 30 min at 20 °C, under agitation. Cells were washed three times with 50 mM glycine in PHEM buffer and after centrifugation were embedded in 12% gelatin in PHEM. The gelatin-embedded blocks were cryo-protected in 2.3 M sucrose in rotating vials at 4 °C (16 h). Samples vitrification was obtained in liquid nitrogen following the plunge and freezing technique[13]. Thin sections (80 nm) were prepared at − 110 °C with a diamond knife (Diatome, Switzerland). Ribbons were picked-up with a drop of 1% (w/v) methylcellulose/1.15 M sucrose in PHEM buffer. Sections were thawed and transferred to Formvar carbon-coated nickel grids.

Immunolabelling was performed using an automated system (Leica microsystems EM IGL). Samples were post-fixed with 2% glutaraldehyde in PBS, pH 7.4, for 5 min and finally washed (three times in PBS, pH 7.4, for 2 min and six times with deionized water for 2 min). Six nanometers gold conjugate goat anti-rabbit secondary antibodies (Aurion, Wageningen, the Netherlands, catalogue number: 806.011, dilution 1/5) were used to detect PsaA and PetA. Goat anti-rabbit gold ultra-small 1.4 nm secondary antibodies (Aurion, Wageningen, the Netherlands, catalogue number 800.011, dilution 1/20), were used to detect PsbA, PsaC and PsbC and sections were enhanced with silver (Aurion R-Gent SE-EM) for 25 min and again washed on deionized water (six times for 2 min). For observation,

grids were incubated 5 min on 2% uranyl acetate (pH 7.0) and transferred to a mixture of 1.6% methyl cellulose and 0.4% uranyl acetate on ice, the excess of the viscous solution was drained away and the grids were let to dry. Grids were imaged in an electron Tecnai 12 electron microscope (FEI, USA), using an Orius CCD camera (Gatan, USA). Primary antibodies were used at a dilution of 1/50. Gold particle counting for statistical analysis was done manually. First, the total number of labels (11,932) was assessed and then particles were attributed to various compartments. If gold particles were uncertainly located (3,995), they were not considered for further analysis.

**Principal components analysis.** The principal components analysis (PCA) allows reducing the dimensionality[39] detecting possible groupings in a given data set[40]. We performed PCA on our immunolabelling data considering four possible subcellular compartments for the antibodies against PSI, PSII and the cytochrome $b_6 f$ complex: the internal (core) and external (peripheral) thylakoid membranes, as well as the pyrenoid and the envelope, to account for possible aspecific labelling. This led to a 4-dimension localization space (core, peripheral, pyrenoid and envelope) of 258 images from four independent cultures, where values represent the number of immunolabelling in a given localization. For data analysis, we first normalized the localization space of each of the 258 images. To do so, for each localization, we calculated the number of gold particles for a given image minus the average number of particles in that localization (for example, for a $x(i,j)$, we obtained $x1(i,j) = x(i,j) − mean(column\ j)$. This value was then normalized by the s.d. in the same localization (for example, $x1(i,j)/s.d.(column\ j)$ in the above considered case).

To represent the distribution of these normalized dimensional data for the 258 images, the direction (a four-dimensional vector) giving the largest possible variance of the distribution (that is, accounts for as much of the variability in the data as possible) was selected as the direction for the first principal component. Then, the direction (another four-dimensional vector) orthogonal to the previous one(s) giving the largest possible variance of the distribution was selected as the direction for the second principal component. The repetition of this procedure automatically selects vectors representing the scatter of the distribution from major ones to minor ones. Based on singular value decomposition, PCA is a principal axis rotation of the original variables that preserves the variation in the data. Therefore, the total variance of the original variables is equal to the total variance of the principal components. The principal component coefficients correspond to the percentage of explained variance. All statistical analysis was done with the R software[41].

**Logistic regression.** Logistic regression was used to describe data and to explain the relationship between one dependent binary variable and one or more independent variables. The two major assumptions are: (i) that the outcome must be discrete, that is, the dependent variable should be dichotomous in nature and (ii) there should be no high intercorrelations (as demonstrated[42], the assumption is met for values less than 0.9) among the predictors.

We use a dose–response relationship model where the predictors are the multiple continuous variables, that is, the number of immunolabelling in the different localizations (core, peripheral, pyrenoid and envelope). Since probabilities have a limited range and regression models could predict off-scale values below zero or above 1, it makes more sense to model the probabilities of getting a given antibody on a transformed scale; this is what is done in logistic regression analysis[43]. A linear model for transformed probabilities can be set up as $logit(p) = \alpha_0 + \alpha_1 x_1 + \cdots + \alpha_k x_k$ in which $\log it(p) = \log((p/1 − p))$ is the log odds. Each $x_i$ is the number of gold beads in the localization $i$ and statistics about the coefficients $\alpha_i$ will provide insight about the impact of the localization $I$ on the probability to get a given antibody. The analysis of deviance table and the Akaike information criterion allows the identification of the relevant predictors[44].

The table of correlations shows that there are no strong intercorrelations between the variables (Supplementary Note 1). Starting from a complete model (Supplementary Note 1) and based on the variable coefficients $P$ values ($Pr(>|z|)$), we see that we can recursively delete the two variables envelope, env. and pyrenoid, pyr. without significantly reducing the Akaike information criterion[45], which is a common measure of the relative quality of a statistical model for a given set of data. The final model demonstrates that the relevant variables to predict the antibody are the number of immunolabelling in core ($P$ value $= 2\ e − 03$) and peripheral, per. ($P$ value $< 8\ e − 8$) area. Bootstrap procedure allows the evaluation of the average percentage of wrong prediction (18%, Supplementary Table 2).

**FIB-SEM and 3D reconstruction.** *P. tricornutum* cells were fixed in 0.1 M cacodylate buffer (Sigma-Aldrich), pH 7.4, containing 2.5% glutaraldehyde (TAAB), 2% formaldehyde (Polysciences) for 1 h at 20 °C and prepared according to a modified protocol from (https://ncmir.ucsd.edu/sbem-protocol). FIB tomography was performed with a Zeiss NVision 40 dual-beam microscope. In this technique, the Durcupan (Sigma-Aldrich) resin-embedded cells of *P. tricornutum* were cut in cross-section, slice by slice, with a Ga + ion beam (of 700 nA at 30 kV). After a thin slice was removed with the ion beam, the newly exposed surface was imaged in SEM at 5 kV using the in-column EsB backscatter detector. For each slice, a thickness of 4 nm was removed, and the SEM images were recorded with a

pixel size of 4 nm. The image stack was then registered by cross-correlation using the StackReg plugin in the Fiji software.

For 3D reconstruction, a stack of 600 images was analysed with FIJI ImageJ software and projected in three-dimension ($x,y,z$ axis) using the AVIZO (FEI, USA) and CHIMERA softwares (https://www.cgl.ucsf.edu/chimera/, UCSF, USA). Experiments were also performed at higher resolution (voxel size 2 nm, Supplementary movie 2). However, no significant improvement of the resolution was observed in these conditions. This likely stems from the fact that under these imaging conditions, the recorded backscatter signals emerge primarily from an area < 5 nm across and < 10 nm thick, setting an empirical lower limit for pixel size and slicethickness[46]. Moreover, the higher electron dose per surface unit might also enhance electron beam fluctuations during the acquisition and/or thermal damages to the sample, thus further limiting the imaging resolution.

**Data availability.** The authors declare that all data supporting the findings of this study are available within the manuscript and its supplementary files or are available from the corresponding authors on request.

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

## Acknowledgements

The authors are grateful to Pierre Joliot (Institut de Biologie Physico Chimique, Paris, France), Arthur Grossman (The Carnegie Institution, Stanford, USA) and Chris Bowler (Ecole Normale Supérieure, Paris, France) for critically reading the manuscript. We thank the Marie Curie Initial Training Network Accliphot (FP7-PEOPLE-2012-ITN; 316427 to S.F., G.F.), the HFSP (HFSP0052 to G.F.), GRAL (ANR-10-Labx-49-01 to B.G., C.M., L.F.E., D.P., C.B., G.S.), the DRF impulsion FIB-Bio program (to P.-H.J., B.G., C.M., L.F.E., D.F., G.S., G.F.), the University of Konstanz (to A.S., C.R.B., P.G.K.) and a stipend from the Graduate School of Chemical Biology (KoRS-CB to A.S.). This work used platforms from ScopeM (ETH Zurich) and the Grenoble Instruct Centre (ISBG: UMS 3518 CNRS-CEA-UJF-EMBL) with support from FRISBI (ANR-10-INSB-05-02) within the Grenoble Partnership for Structural Biology (PSB).

## Author contributions

B.B., A.S., C.R.B., E.M., P.G.K., D.P., S.Z., C.B., G.S., D.F. and G.F. designed the study. S.F., P.-H.J., B.B., B.G., L.F.E., C.M., O.B., S.E., C.B., D.F., and G.F. performed the experiments (S.F.: chloroplast purification, biochemical analyses, spectroscopy, immunolabelling and tomography; P.-H.J.: tomography; B.B.: spectroscopy; B.G.: immunolabelling; L.F.E. tomography; C.M.: immunolabelling; O.B.: PCA analysis; S.E.: tomography; C.B.: biochemical analyses; D.F. immunolabelling and tomography; G.F.: spectroscopy). S.F. B.B., B.G., L.F.E. C.M., O.B. E.M., D.P., S.Z., C.B., G.S., D.F. and G.F. analysed the data. C.B., S.Z., D.F. and G.F. wrote the manuscript, and all authors revised and approved the manuscript.

## Additional information

**Competing interests:** The authors declare no competing financial interests.

