## [Peer Review File · Nature Communications]

Reviewers' comments:

Reviewer #1 (Remarks to the Author):

The presented work by Flori and co-workers describes a functional and structural analysis of thylakoid membrane organisation in the marine diatom *Phaeodactylum*. The authors use absorption spectroscopy measurements to assess the photosynthetic electron flow in the presence and absence of DCMU, which poisons PSII. If PSI and PSII are in close physical contact, i.e. randomly distributed in the membrane as generally thought in diatoms then one would expect the PSI antenna cross-section to increase as PSII is poisoned due to more efficient spillover. In contrast to this prediction very little spillover is observed, suggesting spatial operation of PSI and PSII. This idea is then tested structural by immunogold labelling of the cryo sections. The results suggest PSI is preferentially found in stroma exposed thylakoids whereas PSII is located in the core membranes; the fact that PSII is less accessible to digitonin than PSI supports this view. 3D reconstruction of the cryosections suggests that the two domains are linked via membrane bridges, which could facilitate diffusion of cyt c6 from the PSII region to PSI.

The statistical analysis of the gold labelling and the electron transfer experiments are all performed correctly and the results are certainly novel. The work touches on the structural complexities in membrane architecture employed by nature to solve diffusion problems and is therefore of interest to a wide audience.

I have two slight reservations that I believe warrant refinement, but which can be easily addressed by the authors in short time:

1) the suggested use of the membrane bridges is to facilitate diffusion between PSI and PSII regions, to this end it would be useful to know whether cytochrome b6f complexes exist in the PSII domains (a missing immunolabeling experiment). If they are excluded their role as highways for cyt c6 diffusion is in doubt.

2) the correspondence between the 3D images shown in figure 5 and the model drawn in 5d are unclear to my eyes. For instance the membrane bridges do not appear in the model, while it is understood how the bridges span the PSI and PSII membranes in 5b and c. The picture needs to be made clearer, could the membranes and connections be outlined?

Reviewer #2 (Remarks to the Author):

This interesting manuscript reports on the architecture of the thylakoid membrane system in diatoms which form an important part of the marine phytoplankton. In contrast to land plants, the diatom chloroplast derived from red algae via secondary endosymbiosis. As a consequence, the thylakoid system exhibits no differentiation into grana stacks and stromal lamellae. Such thylakoid sub-domains of land plants enable lateral heterogeneity of PSII

and PSI accumulating in grana and stroma lamellae regions, respectively. Since this morphological organization avoids energy spillover from PSII to PSI, the questions arise if and how diatoms organize their PSs.

This provided the entry point for the work of the authors who clearly showed a very limited spillover of energy after inhibiting PSII by DCMU (Figure 1). Consistently, diffusion domains of PSs were verified via measuring redox kinetics (Figure 3). This strongly suggests that a functional separation between PSII and PSI exists despite the absence of specialized grana regions.

Next the authors correlated this data with the thylakoid sub-structure of diatoms. By applying EM-based immunogold-labeling of PSII and PSI subunits, they detected PSII preferentially in the central core thylakoid membranes whereas PSI located to the external thylakoids of the usual bundle of three thylakoid lobes, i.e., six thylakoid membranes (Figures 2 and 5d). The statistical evaluation appears clear. In addition, a biochemical approach revealed a higher accessibility of PSI as compared to PSII to mild detergent treatment supporting the idea of a peripherally localized PSI (Supp. Figure 4). Nonetheless, this part of the work requires additional controls and/or a partly different experimental set-up. This appears important, because the localization of PSII and PSI brings together the physiological and morphological aspects of the work and thus forms the heart of the authors' model.

In immunogold labeling experiments the quality of the antibody is critical. Therefore, several controls are usually included into the assay. First, a sample in the absence of the first antibody should be tested for cross-reactivity of the secondary antibody. Secondly, the specificity of the first antibody could be verified by analyzing the pre-immune serum (if available) as a negative control. Since no PSII or PSI knockout mutants for *Phaeodactylum tricorutum* are available (which would allow to test for antiserum specificity), I would suggest to analyze antibodies for a different subunit of either PSII or PSI in parallel. This should result in similar PS distributions and thereby strongly substantiate the conclusions.

Additionally, a PSI antibody different from PsaC should be used in particular for the biochemical solubilization experiment. PsaC is an extrinsic subunit of PSI whereas D1 represents an intrinsic subunit of PSII. Thus, an intrinsic PSI subunit like PsaA or PsaB would be much more appropriate to be followed.

Before publishing some minor points should also be considered:

- Labeling in Figure 2a (PS2 + PsaA = D1) and 2b (PS1 + PsaC) is confusing. α D1 (or α PsaA) and α PsaC would be better.
- In the text on page 5, lines 97, 98, it is incorrectly referred to Figure 2.
- In Figures 4 and 5, scale bars are lacking.
- Figure 4 appears a bit distracting because it focusses on aspects outside the chloroplast. It might be combined with Figure 5.
- Original data sets of representative sections used for the 3D reconstruction in Figure 5b and 5c should be shown aside. Moreover, Figures 5b and 5c require better labeling. Where are the lumen and stromal intermembrane regions located? Do the color codes correspond to Figure 5d?
- It would be very helpful for the reader, if the connections seen in Figure 5b,c could be integrated into Figure 5d. Moreover, it might be helpful for fast recognition of lumen areas if these would be color-coded, too.

- Which antibody was used in Supp. Figure 5?

Reviewer #3 (Remarks to the Author):

In this manuscript, Flori et al. use a combination of electron microscopy, biochemistry, and functional assays to gain insights into the molecular organization and 3D architecture of diatom thylakoid membranes. This work makes two main discoveries:

1) The photosystem complexes PSI and PSII are segregated to different thylakoid membrane sub-domains, with PSI on the stroma-facing "peripheral" membranes on the outsides of the thylakoid stacks and PSII on the "core" membranes within the stacks. This conclusion is based on live-cell spectroscopy (Fig. 1), immuno-em (Fig. 2), and detergent solubilization (Fig. S4). While more direct visualization via freeze fracture / deep etch or cryo-electron tomography would have aided the conclusion, the combination of these three approaches does provide fairly compelling evidence for compartmentalization. This segregated localization of PSI and PSII means that diatom stacks are similar in their molecular organization to higher plant grana (where the phenomenon is called lateral heterogeneity) and stacked thylakoids of the alga *Chlamydomonas* (see old freeze fracture papers by Olive, Wollman, and Recouvreur). I am not versed enough in the diatom literature to know whether this conclusion is novel. If it is, then this is a very important finding that advances our understanding of diatom photosynthesis. The similarity to other photosynthetic organisms implies that comparable regulatory mechanisms may be at work.

2) The authors describe what appear to be arrays of single "linker" thylakoids that join adjacent thylakoid stacks (Fig. 5), facilitating the rapid diffusion of electron carriers and redox carriers (Fig. 3). To my knowledge, this is a unique structure of diatom thylakoid networks. It is also quite an important discovery, as it provides a mechanism for optimizing photosynthetic efficiency in the changing light conditions of the ocean.

In summary, this work is impactful because it paints a clearer picture of how thylakoid architecture orchestrates photosynthesis in diatoms, some of the most ecologically important autotrophs on the planet. However, there are several significant issues that must be addressed prior to publication. Fortunately, most of these issues can be resolved through revision of the manuscript and figures:

1) Throughout the manuscript, the authors compare their diatom observations to published data on higher plants. However, the same comparisons should also be made to data on *Chlamydomonas*, the most thoroughly studied alga, which is evolutionarily closer to diatoms. The authors should comment on which features they believe are distinct adaptations of diatoms, and which may rather be characteristics shared by different types of algae. Here are some examples:

a. Line 131: "Thus, the compartmentalisation of PS1 and PS2 in different thylakoid domains

also generates diffusion domains in *P. tricornutum*, similar to plants. However, their equilibration time, 10 ms (corresponding to 100 electrons s⁻¹), is much faster than in plants, ~150 ms (corresponding to ~ 7 electrons s⁻¹).

How do diffusion rates compare to *Chlamydomonas*, which has a more similar thylakoid architecture to diatoms than to plants? Is the rapid equilibration time a result of not having grana (shared by all algae), or is it a distinct property of diatoms?

b. The segregation of PSI from PSII (with PSI in the “peripheral” membranes and PSII in the “core” membranes) is also seen in the loose thylakoid stacks of *Chlamydomonas* (Wollman et al. 1980, JCB). Thus, perhaps this compartmentalization is a more general property of all stacked thylakoid membranes, not just the grana systems of higher plants.

c. Line 144: “we observed regions where thylakoid membranes are apparently interconnected or where they abruptly “disappear” in cross sections”

Both interconnecting thylakoid stacks and the termination (“disappearance”) of single thylakoids from a stack and have been shown quite clearly in cryo-tomograms of *Chlamydomonas* (Engel et al. 2015, eLife). Thus, these are architectural features that are likely common to different types of algae with stacked thylakoids but no grana.

d. One difference between *Chlamydomonas* and diatoms that should be mentioned is that while diatoms always have 3 thylakoids in a stacks, the number of thylakoids in a *Chlamydomonas* stack can vary from 2 to 15, with a median of 3 (Engel et al. 2015, eLife; Polukhina et al. 2016, Plant Phys). Despite these differences, the thylakoid stacks of these two organisms appear to share several properties (as discussed above).

e. The measurement of thylakoid lumen width should also be compared to *Chlamydomonas*. See the detailed explanation in point 2 directly below.

2) Line 135: “To explain the different equilibration time of diatoms and plants, we reexamined the EM pictures of samples prepared with the Tokuyasu protocol. By preserving the membrane structures, this technique allowed additional features of the *P. tricornutum* thylakoids to be observed. First, we found that the size of the *P. tricornutum* lumen (7.1 ± 1.5 nm, average of 100 estimates from 67 different preparations) was larger than that of plants (4.5 nm), consistent with the expected loose membrane stacking in diatoms. This size is double the size of the soluble electron carrier cyt c6 (~ 33 x 23 Å), while in plants the lumen size is equivalent to that of the soluble redox carrier plastocyanin. The larger lumen size in *P. tricornutum* should facilitate the diffusion of the soluble carriers.”

These conclusions are misleading and require changes:

a. In higher plants, thylakoid lumen width is dynamic and swells from 4.5 nm in dark-adapted grana (Kirchhoff et al. 2011, PNAS; Daum et al. 2010, Plant Cell) to 9 nm in plants that have been adapted to the light (Kirchhoff et al. 2011). In other words, when the light reactions of photosynthesis are active, the plant thylakoid lumen is 9 nm. Thus, it is not

true that diatoms have more luminal space for the diffusion of soluble carriers.

b. When grown in the light, *Chlamydomonas* also has a thylakoid lumen width of 9nm (Engel et al. 2015). It is currently untested whether this width shrinks in the dark, like higher plants. Regardless, *Chlamydomonas* also has sufficient luminal space for the diffusion of soluble carriers.

c. Later in the manuscript, the Kirchhoff et al. observation is finally cited. However, it is done in a confusing and conflicting way. Line 178: "Finally, a size of the lumen larger than the size of the soluble electron carrier plus the presence of interlinking thylakoids should facilitate diffusion of cyt c6, thus allowing fast redox equilibration between the photosynthetic complexes in the diatom, unlike plants. Consistent with this idea, lumen widening has been observed in plants exposed to high light to facilitate electron flow and prevent photodamage in plants"

These two sentences directly conflict with each other. The authors say that the diatom lumen is wider, thus enabling rapid diffusion and equilibrium of soluble carriers, unlike in plants. And then they say that the expansion of the plant thylakoid lumen in light is consistent with this idea. However, it is not. As the plant thylakoid lumen is at least as wide as the diatom lumen during light-driven photosynthesis (or maybe even wider: 9 nm vs. 7 nm), a difference in luminal width cannot explain the difference in redox equilibrium rates between diatoms and plants. In contrast, it does seem plausible that the increased interconnectivity provided by "linking" thylakoids improves diffusion in diatoms.

d. Given the published luminal widths from Kirchhoff et al., Daum et al., and Engel et al. (each of which should be clearly cited and described in the manuscript), how do the authors interpret their measurements in diatoms? Importantly, what were the photosynthetic conditions of the cells when they were fixed in aldehyde for electron microscopy? The 7 nm diatom measurement falls in between the 9 nm high-light and 4.5 nm dark luminal widths reported for plants and *Chlamydomonas*. Were the diatoms grown in medium or low light (please contextualize $20 \mu\text{mol photon m}^{-2}\text{s}^{-1}$ with 12 hr light/dark cycle), and thus were the cells performing an intermediate level of photosynthesis? This might make sense with a 7 nm width, assuming thylakoid dynamics are consistent between plants, *Chlamydomonas*, and diatoms. Additional dark and high-light measurements of the diatom thylakoids would be very informative. However, if those experiments cannot be accomplished for this publication, the existing data should be put in the proper context.

3) In Figs. 4 and 5, real images from the FIB/SEM 3D volumes must accompany the segmentations that are shown. While segmentations are helpful for interpreting 3D morphology, they are only an interpretation. It is absolutely essential that features in the actual micrographs are displayed for the readers. 3D volumes can be rotated and sliced in software such as IMOD's 3dmod slicer window to show precisely the features of interest in the desired orientation.

a. In Fig. 4, there should be a new first panel that shows a longitudinal end-to-end overview slice of the cell, rotated from the real FIB/SEM data so that the whole cell can be seen in

one image. This image should be in the same orientation of the current panel A, which will then become panel B.

b. In Fig. 4, a FIB/SEM close-up image of the "evagination" contact between the chloroplast and nucleus is needed to accompany panel D. Only by seeing the real data can we judge the nature of the membrane contact. This contact is also not very evident in the view of the segmentation that is shown. A better view should be displayed, perhaps with an arrow pointing to the contact.

c. In Fig. 5, two FIB/SEM close-up images are needed to accompany the segmentations of "linking thylakoids" in panels B and C. The black and red arrows pointing to thylakoids and plastoglobules in panel C should also be indicated on the FIB/SEM micrographs. This is a potentially novel structure, so it is very important that the readers are shown what it looks like in the actual EM data.

Minor points that should also be addressed:

1) The authors use the names PS1 and PS2, but I believe the standard nomenclature is PSI and PSII. This should be changed throughout the manuscript.

2) Line 94: "we immunolocalized the two PSs in cells prepared using the Tokuyasu protocol, an optimal method to preserve membrane structures for electron microscopy (EM) imaging"

As stated in the methods, the Tokuyasu protocol involves standard aldehyde fixation of the biological material, which is known to cause deformations of membranes. This is certainly not an "optimal" method to preserve membranes. While the optimal approach would be FIB-thinning of vitreous frozen-hydrated cells, the best option for conventional resin-embedded EM would be high pressure freezing followed by freeze-substitution. I do not believe that the aldehyde fixation used by the authors invalidates any of their results, but the wording should be changed to: "we immunolocalized the two PSs in cells prepared for electron microscopy (EM) using the Tokuyasu protocol, a cryo-sectioning method that improves the preservation of membranes structures".

3) The biochemical prep in Fig. S4 is one of the major lines of evidence used by the authors to conclude that PSI and PSII are partitioned to specific thylakoid compartments. Thus, this data should not be in the supplement, but rather should be added to Fig. 2 to accompany the immuno-EM localization.

4) The labels "d" and "f" in Fig. 1, "b" and "d" in Fig. 3, and "c" in Fig. 5 should be moved to the upper left of the corresponding panels. This is the standard label position for scientific figures.

5) In Fig. 1A, the label "common antenna" should be changed to "shared antenna". The word "common" has a common secondary meaning, which I just used in this sentence.

6) When Fig. 2 is printed, the dark blue color denoting the envelope is very similar to the purple color showing the "core" membranes. The dark blue should be changed to light blue or magenta.

7) In Fig. 3, the overlapping shape symbols make the plots in panels B, C, and D quite hard to read. A less cluttered plotting method with colored or patterned lines might be preferable.

8) The phrase "lateral heterogeneity" should be used at least once in the manuscript to describe the segregated localization of PSI and PSII to different thylakoid regions.

9) The English prose in this manuscript would benefit from a good polishing. I list some edits below, but this list is not exhaustive:

a. Remove "i.e." from lines 57, 59, 116, 128, 130, and 173. Each of these sentences makes sense and is easier to read without "id est" getting in the way.

b. Throughout the manuscript, "via" should not be italicized. Although from Latin origin, it is used in standard English.

c. Line 58: "Primary plastids comprise differentiated thylakoid domains"
Change to: "Primary plastids contain differentiated thylakoid domains"

d. Line 73: "a sophisticated thylakoid membrane network that orchestrates photosynthetic light absorption and utilisation via subtle subthylakoid segregation of the PSs."
Change to: "a sophisticated thylakoid membrane network that orchestrates photosynthetic light absorption and utilisation via segregation of the PSs to specific thylakoid subdomains."

e. Line 78: "thereby favouring energy spillover via physical contacts."
Change to: "thereby favouring energy spillover via physical contacts between the complexes."

f. Line 82: "This phenomenon has been documented in PS2-poisoned red algae and interpreted as a signature of spillover in these algae, considered the ancestors of secondary plastids."
Change to: "In red algae, considered the ancestors of secondary plastids, this phenomenon was observed upon poisoning of PSII and was interpreted as a signature of spillover."

g. Line 91: "Thus, either (i) physical barriers-created by lipid/biochemical surroundings-prevent energy exchange"
Change to: "Thus, either (i) lipid or biochemical barriers prevent energy exchange"

h. Line 102: "In plant thylakoids (see e.g.12)"
Change to: "In plant thylakoids¹²"

i. Line 116: "an 'equilibration plot' (Fig. 3), i.e., the relationship"

Change to: "an 'equilibration plot' (Fig. 3), which shows the relationship"

j. Line 149: "focused ion-beam"

Change to: "focused ion beam"

k. Line 150: "We identified the organelles and their interactions"

Change to: "By segmenting the 3D volume, we identified the organelles and their contacts"

l. Line 158: "A focus on the 3D structure of the photosynthetic membranes (Fig. 5a-c) confirmed the presence of parallel layers of thylakoids in the plastid but also revealed the presence of thylakoids (Fig. 5b,c, green) connecting the different thylakoid layers"

Change to: "The 3D structure of the photosynthetic membranes (Fig. 5a-c) confirmed the presence of parallel layers of stacked thylakoids, but also revealed the presence of single thylakoids (Fig. 5b,c, green) connecting the different thylakoid stacks"

m. Line 163: "Our 3D reconstruction of the *P. tricornutum* plastid with FIB-SEM discloses"

Change to: "Our 3D FIB-SEM reconstruction of the *P. tricornutum* plastid reveals"

n. Line 167: "Subtle compartmentalisation"

Change to: "compartmentalisation". There is nothing in the data to indicate that it is subtle.

o. Line 173: "state transitions... does not exist in diatoms"

Change to: "state transitions... do not exist in diatoms"

And is this a certainty? It may be better to say: "state transitions... are believed to not exist in diatoms"

Reviewer #4 (Remarks to the Author):

The manuscript by Flori et al. deals with one of the longstanding questions in research on eukaryotic algae belonging to the ecologically extremely important group of secondary endosymbionts, the diatoms. In contrast to higher plants thylakoids are not separated in grana and stroma thylakoids, but organised in lamellae of mostly three thylakoids each spanning the whole plastid. This poses the question of energetic separation of photosystem II and I and the feasibility of electron transport in such an organisation. Thus, the manuscript deals with a very important question in photosynthetic research on this important group of organisms.

The manuscript is tackling three main research questions, i) is there spill-over (direct energy transfer) from PSII to PSI, ii) are the photosystems equally distributed over the six membranes of the lamellae, and iii) are there connections between single thylakoids in a/between lamellae enabling electron transfer between them. First I would like to discuss the novelty of the findings:

i) Spill-over was not explicitly addressed in diatoms before. However, there are several studies on excitation energy transfer *in vivo*, and neither found strong indications for a huge amount of spill-over. Miloslavina et al. *Biochim. Biophys. Acta* 1787(10), 1189-1197 (2009) discuss only one of their minor components in this context, as do Yokono et al. (Ref 10 in

the manuscript), whereas Chukhutsina et al. *Biochim. Biophys. Acta* 1827(1), 10-18 (2013) did experiments with and without DCMU like done here and the lifetime of PSI did not change. Although this was not explicitly discussed as a lack of spill-over there is no other explanation for it. However, nobody measured spill-over in diatoms on the basis of electron transport before as done here.

ii) The results of Ref 7 are not cited entirely: this study was the first to detect an uneven distribution of PSI with a preference for the outer membranes of a lamellae using immunolabelling in electron microscopy, but for whatever reason authors do not refer to this major result of the paper. PSII was never tested, and although the hypothesis of PSII being located in the inner membranes is not new, nobody has ever proven it experimentally before.

iii) so far only transmission electron microscopy data obtained after thin sectioning exist about the thylakoid structure of diatoms. Thus, every higher resolution data or 3D reconstruction is of great value.

However, my main concern is about the interpretation of the FIB-SEM data on thylakoid 3D structure. According to Material and Methods the data were acquired with a pixel size of 4x4 nm, and removal of about 4 nm for each slice in z-direction, ending up in roughly 4x4x4 nm voxels. A membrane has a thickness of 4-5 nm. By imaging using the described voxel size, single membranes are not to be distinguished. Even for a whole thylakoid (about 10 nm for the two membranes, plus around 7 nm for the lumen) the resolution is beyond its limits with only four voxels in z-direction. As a consequence, Fig. 5b and c show noisy surfaces of unknown attribution, but shed no light on the interesting question of thylakoid 3D structure in diatoms. Also for the really nice TEM images shown here, slices were 80 nm, so no distinction between overlays and crossings can be made (Fig. S5) due to the sample thickness. For both methods cells were fixed with glutaraldehyde, which was often discussed as critical for measuring exact membrane distances. Also the finding that there are not always three thylakoids per lamellae, but sometimes 'suddenly' four or only two is old (Ref 5 and 7). Some further major concerns can be found below.

Thus, novelty is limited and authors should be more careful in the description of whether data they present are confirming former publications (no spill-over, PSI localisation, nucleus-chloroplast interaction) or new (PSII localisation). Most importantly they should limit their interpretation of electron microscopy data to the resolution obtained or better, acquire higher resolution data.

Further major points

1) The nice spectroscopic analysis of P700 and cyt oxidation shows no deviation between curves measured with or without DCMU (+HA) in Fig. 1 of the main text. However, in the supplementary file data obtained using different light intensities are shown. In Fig. S2 it is obvious that the deviation between data obtained with and without DCMU+HA is strongly light intensity dependent, with a maximum of about 20% at the highest intensity. This needs explanation and discussion, since 20% is far from negligible or 'limited' (line 90). In addition there is a contradiction between Fig 1 and Fig S2f: according to the legends the same light intensity was used but graphs differ tremendously, which light intensity was used in Fig. 1?

2) Due to the spectroscopic overlap it is only possible to probe the redox state of cyt c and

cyt f simultaneously, as rightly stated in Material and Methods. However, in the Results authors always use cyt c, which is misleading. Please use either cyt or [cyt c + cyt f]. This is especially important in the sections about diffusion barriers, since the calculations cannot discriminate between cytf/cyt c and cyt c/PSI electron transfer. In addition, please mention the light intensity used in the legend of Fig. 3

3) Estimations of the luminal space or the distance between thylakoids from EM data are always hampered and were often criticised in cases where fixatives were used like in this study. In Ref 7 already 6 nm were reported. SANS studies like those of Nagy et al. Biochem. J. 436(2), 225-230 (2011) revealed 17 nm for a whole thylakoid in vivo. If taking the usual estimate for membrane thickness (4-5 nm) this would leave maximal 9 nm for both the space between thylakoids and the lumen. Those values should at least be discussed.

4) FIB-SEM, line 153-157: The contact between nucleus and chloroplast was to be expected, since the outer membrane of the chloroplast envelope in heteroconts is in connection with the nuclear ER due to their evolutionary history. These features are quite different to the cited plant stromules. For Fig. 5 the scale bar is missing and it would be nice to know what and why certain areas were shaded green and purple in b and c.

5) Material and Methods suffers from lengthy descriptions of e.g. standard statistical methods (PCA), but many important details are missing:

When was DCMU added in the different experiments?

It seems that for cytochromes extinction coefficients were used, whereas PSI oxidation is plotted on a relative scale, please explain/mention.

The rate of generation of electrons by PSII (Results, lines 127 – 134 and Fig. 3) was probably done as in Ref 3, i.e. using the dark relaxation. This should be mentioned.

Intactness of chloroplasts is given as 70%. The method used tests the intactness of the envelope (penetration of ferricyanide). For an estimate of overall intactness the absolute O₂ evolution rates have to be given in comparison to the rates seen in cells.

Line 280/281: are these the concentrations of the detergent solutions or the final concentrations? If the former is true, how much was added? Which method was employed to measure protein concentration? Which gels were used for the final analysis (7% or 13%)?

Which secondary antibody was used for the gold-labelling, and, most importantly: what was the size of the gold label?

Authors did a sophisticated statistical analysis, but how were the images analysed in order to acquire the basic data set? Antibody labelling is often seen in areas belonging to the lumen of the outmost thylakoid (e.g. Fig. 2c, lower right corner). Were these labels neglected, counted to outer membranes or core membranes?

Fig S3: are the scale bars correct? They imply that a factor of 2.5 is in between both pictures – the difference in size of e.g. thylakoid lamellae looks less

Minor points

Line 67/68: deviations from the 3 thylakoid per lamellae scheme were already reported by Refs 5 and 7. Thus the sentence would be more informative if rephrased to 'loose stack of mostly three thylakoids, in some cases two or four, with a few...' and both references cited, since also anastomoses were detected by Ref 7

Line 78: Ref 7 does not show a random distribution of PSI, on the contrary (see above). Only FCPs were randomly distributed

Line 97: the preferential localisation of PSI in the outer membranes requires a reference to Ref 7 ('as shown before')

Line 97/98: in line 97 is should read Fig 2b, c, whereas in line 98 Fig 2a, c is correct

Line 146: anastomoses are not 'unexpected', see for example Ref 5 and 7

Line 166: The paper cited (Ref 18) has nothing to do with diatoms and their thylakoid structure

Line 169: nobody measured lipid distribution, so the observations are compatible 'with the hypothesis that the core membranes are enriched in lipids....'

Line 178: there is indeed a group of organisms with similar thylakoid structure (dinoflagellates) where recently a huge spill-over was reported under high light conditions (Slavov et al Biochim. Biophys. Acta 1857(6), 840-847 (2016))

Line 211: wrong format for multiplication in formula

Line 223: '...reduction rate of this electron donor pool' would be more clear

Line 230: last word 'not'

Line 257: should be in bold

Line 299: what does (10) stand for?

Line 339: was

Line 373: which appendix?

Line 450: light (not fight)

Refs 21 and 22 seemed to be swapped

Answers (in red) to Reviewers' comments (in black):

Reviewer #1 (Remarks to the Author):

The presented work by Flori and co-workers describes a functional and structural analysis of thylakoid membrane organisation in the marine diatom *Phaeodactylum*. The authors use absorption spectroscopy measurements to assess the photosynthetic electron flow in the presence and absence of DCMU, which poisons PSII. If PSI and PSII are in close physical contact, i.e. randomly distributed in the membrane as generally thought in diatoms then one would expect the PSI antenna cross-section to increase as PSII is poisoned due to more efficient spillover. In contrast to this prediction very little spillover is observed, suggesting spatial operation of PSI and PSII. This idea is then tested structural by immunogold labelling of the cryo sections. The results suggest PSI is preferentially found in stroma exposed thylakoids whereas PSII is located in the core membranes; the fact that PSII is less accessible to digitonin than PSI supports this view. 3D reconstruction of the cryosections suggests that the two domains are linked via membrane bridges, which could facilitate diffusion of cyt c6 from the PSII region to PSI.

The statistical analysis of the gold labelling and the electron transfer experiments are all performed correctly and the results are certainly novel. The work touches on the structural complexities in membrane architecture employed by nature to solve diffusion problems and is therefore of interest to a wide audience.

We would like to thank the reviewer for the nice summary of our work and the encouraging feedback.

I have two slight reservations that I believe warrant refinement, but which can be easily addressed by the authors in short time:

1) The suggested use of the membrane bridges is to facilitate diffusion between PSI and PSII regions, to this end it would be useful to know whether cytochrome *b₆f* complexes exist in the PSII domains (a missing immunolabeling experiment). If they are excluded, their role as highways for cyt c6 diffusion is in doubt.

This is an excellent suggestion. To address the localisation of the cytochrome *b₆f* complex, immunolabeling experiments have been repeated using an antibody against the cytochrome f subunit (PetA) of this complex. As shown in the new Fig. 2b, cyt *b₆f* localises in the stroma-exposed membranes in cells prepared using the Tokuyasu protocol, similar to PSI (Fig. 2c). This localisation is confirmed by biochemical evidence with cyt *b₆f*, as PSI, being more accessible to mild detergents than PSII (Fig. 2g).

2) the correspondence between the 3D images shown in figure 5 and the model drawn in 5d are unclear to my eyes. For instance the membrane bridges do not appear in the model, while it is understand how the bridges span the PSI and PSII membranes in 5b and c. The picture needs to be made clearer, could the membranes and connections be outlined?

We recognise that the model could be misleading and/or difficult to understand. For this reason, we removed our model from the revised version of Fig. 5. This figure has been further modified (according to the suggestions of other Reviewers) to better highlight the connections between adjacent stacks of thylakoid membranes.

Reviewer #2 (Remarks to the Author):

This interesting manuscript reports on the architecture of the thylakoid membrane system in diatoms which form an important part of the marine phytoplankton. In contrast to land plants, the diatom chloroplast derived from red algae via secondary endosymbiosis. As a consequence, the thylakoid system exhibits no differentiation into grana stacks and stromal lamellae. Such thylakoid sub-domains of land plants enable lateral heterogeneity of PSII and PSI accumulating in grana and stroma lamellae regions, respectively. Since this morphological organization avoids energy spillover from PSII to PSI, the questions arise if and how diatoms organize their PSs. This provided the entry point for the work of the authors who clearly showed a very limited spillover of energy after inhibiting PSII by DCMU (Figure 1). Consistently, diffusion domains of PSs were verified via measuring redox kinetics (Figure 3). This strongly suggests that a functional separation between PSII and PSI exists despite the absence of specialized grana regions. Next the authors correlated this data with the thylakoid sub-structure of diatoms. By applying EM-based immunogold-labeling of PSII and PSI subunits, they detected PSII preferentially in the central core thylakoid membranes whereas PSI located to the external thylakoids of the usual bundle of three thylakoid lobes, i.e., six thylakoid membranes (Figures 2 and 5d). The statistical evaluation appears clear. In addition, a biochemical approach revealed a higher accessibility of PSI as compared to PSII to mild detergent treatment supporting the idea of a peripherally localized PSI (Supp. Figure 4).

Nonetheless, this part of the work requires additional controls and/or a partly different experimental set-up. This appears important, because the localization of PSII and PSI brings together the physiological and morphological aspects of the work and thus forms the heart of the authors' model.

We thank the Reviewer for this good summary of our work and the very useful suggestions to improve it.

- In immunogold labeling experiments the quality of the antibody is critical. Therefore, several controls are usually included into the assay. First, a sample in the absence of the first antibody should be tested for cross-reactivity of the secondary antibody.

This control was of course performed and has now been included in the new version of Fig. S4, where we show lack of labelling in the case of the two Goat anti-rabbit secondary antibodies used in this study (Fig. S4c and d).

- Secondly, the specificity of the first antibody could be verified by analyzing the pre-immune serum (if available) as a negative control.

Unfortunately, no pre-immune serum was available for these antibodies, which were purchased from Agrisera. However the negative controls discussed above testify the specificity of the immunolabelling experiments.

- Since no PSII or PSI knockout mutants for *Phaeodactylum tricornutum* are available (which would allow to test for antiserum specificity), I would suggest to analyze antibodies for a different subunit of either PSII or PSI in parallel. This should result in similar PS distributions and thereby strongly substantiate the conclusions. Additionally, a PSI antibody different from PsaC should be used in particular for the biochemical solubilization experiment. PsaC is an extrinsic subunit of PSI whereas D1 represents an intrinsic subunit of PSII. Thus, an intrinsic PSI subunit like PsaA or PsaB would be much more appropriate to be followed.

As suggested, we have now used two antibodies against PSI (PsaA, Fig. 2a; PsaC, Fig. S4a) and two antibodies against PSII (PsbA, Fig. 2b PsbC, Fig. S4b) to address the localisation of both PSs. Unfortunately, this was not possible in the case of *cyt b₆f*, because only an antibody raised against the PetA (*cyt f*) subunit cross reacted with the *b₆f* complex of *P. tricornutum* in our hands.

Before publishing some minor points should also be considered:

Before publishing some minor points should also be considered: - Labeling in Figure 2a (PS2 + PsbA = D1) and 2b (PS1 + PsaC) is confusing. α D1 (or α PsbA) and α PsaC would be better.

This point has been modified in the revised version of the manuscript.

In the text on page 5, lines 97, 98, it is incorrectly referred to Figure 2.

The whole text has been revised to eliminate these mistakes.

- In Figures 4 and 5, scale bars are lacking.

Scale bars have been added to all figure panels.

- Figure 4 appears a bit distracting because it focusses on aspects outside the chloroplast. It might be combined with Figure 5.

Based on this comments and the comments of the other reviewers, we have substantially modified both Fig. 4 and Fig. 5. Considering these revisions, it is difficult to merge them (Fig. 4 and Fig. 5 contain 7 pictures each). However, the new version of Fig. 4 should convey a more precise message (see also the answer to the other Reviewers below).

- Original data sets of representative sections used for the 3D reconstruction in Figure 5b and 5c should be shown aside.

- Moreover, Figures 5b and 5c require better labeling. Where are the lumen and stromal intermembrane regions located?

-Do the color codes correspond to Figure 5d?

-It would be very helpful for the reader, if the connections seen in Figure 5b,c could be integrated into Figure 5d.

-Moreover, it might be helpful for fast recognition of lumen areas if these would be color-coded, too.

The new Fig. 4 and Fig. 5 present representative sections used for the 3D reconstruction. Moreover, as already discussed above in the case of Reviewer 1, we decided to remove the model presented in the previous Fig. 5d, as it could be misleading.

- Which antibody was used in Supp. Figure 5?

In this figure, an antibody against PsbA (D1) was used. This is now stated in the figure legend.

Reviewer #3 (Remarks to the Author):

In this manuscript, Flori et al. use a combination of electron microscopy, biochemistry, and functional assays to gain insights into the molecular organization and 3D architecture of diatom thylakoid membranes. This work makes two main discoveries:

1) The photosystem complexes PSI and PSII are segregated to different thylakoid membrane sub-domains, with PSI on the stroma-facing “peripheral” membranes on the outsides of the thylakoid stacks and PSII on the “core” membranes within the stacks. This conclusion is based on live-cell spectroscopy (Fig. 1), immuno-em (Fig. 2), and detergent solubilization (Fig. S4). While more direct visualization via freeze fracture / deep etch or cryo-electron tomography would have aided the conclusion, the combination of these three approaches does provide fairly compelling evidence for compartmentalization. This segregated localization of PSI and PSII means that diatom stacks are similar in their molecular organization to higher plant grana (where the phenomenon is called lateral heterogeneity) and stacked thylakoids of the alga *Chlamydomonas* (see old freeze fracture papers by Olive, Wollman, and Recouvreur). I am not versed enough in the diatom literature to know whether this conclusion is novel. If it is, then this is a very important finding that advances our understanding of diatom photosynthesis. The similarity to other photosynthetic organisms implies that comparable regulatory mechanisms may be at work.

2) The authors describe what appear to be arrays of single “linker” thylakoids that join adjacent thylakoid stacks (Fig.5), facilitating the rapid diffusion of electron carriers and redox carriers (Fig. 3). To my knowledge, this is a unique structure of diatom thylakoid networks. It is also quite an important discovery, as it provides a mechanism for optimizing photosynthetic efficiency in the changing light conditions of the ocean.

In summary, this work is impactful because it paints a clearer picture of how thylakoid architecture orchestrates photosynthesis in diatoms, some of the most ecologically important autotrophs on the planet. However, there are several significant issues that must be address prior to publication. Fortunately, most of these issues can be resolved through revision of the manuscript and figures:

We thank the reviewer for this encouraging feedback and believe that our observations about the organisation of the photosynthetic apparatus in diatoms is indeed novel and important.

1) Throughout the manuscript, the authors compare their diatom observations to published data on higher plants. However, **the same comparisons should also be made to data on *Chlamydomonas***, the most thoroughly studied alga, which is evolutionarily closer to diatoms. The authors should comment on which features they believe are distinct adaptations of diatoms, and which may rather be characteristics shared by different types of algae. Here are some examples:

a. Line 131: “Thus, the compartmentalisation of PS1 and PS2 in different thylakoid domains also generates diffusion domains in *P. tricornutum*, similar to plants. However, their equilibration time, 10 ms (corresponding to 100 electrons s⁻¹), is much faster than in plants, ~150 ms (corresponding to ~ 7 electrons s⁻¹).

How do diffusion rates compare to *Chlamydomonas*, which has a more similar thylakoid architecture to diatoms than to plants? Is the rapid equilibration time a result of not having grana (shared by all algae), or is it a distinct property of diatoms?

To our knowledge, such analysis has not been performed in *Chlamydomonas*, but it has been previously employed in *Acaryochloris marina* (a cyanobacterium living in particular environmental niches). We now quote the corresponding publication (reference 38) in the revised version of the manuscript.

b. The segregation of PSI from PSII (with PSI in the “peripheral” membranes and PSII in the “core” membranes) is also seen in the loose thylakoid stacks of *Chlamydomonas* (Wollman et al. 1980, JCB). Thus, perhaps this compartmentalization is a more general property of all stacked thylakoid membranes, not just the grana systems of higher plants.

c. Line 144: “we observed regions where thylakoid membranes are apparently interconnected or where they abruptly “disappear” in cross sections”

Both interconnecting thylakoid stacks and the termination (“disappearance”) of single thylakoids from a stack and have been shown quite clearly in cryo-tomograms of *Chlamydomonas* (Engel et al. 2015, eLife). Thus, these are architectural features that are likely common to different types of algae with stacked thylakoids but no grana.

d. One difference between *Chlamydomonas* and diatoms that should be mentioned is that while diatoms always have 3 thylakoids in a stacks, the number of thylakoids in a *Chlamydomonas* stack can vary from 2 to 15, with a median of 3 (Engel et al. 2015, eLife; Polukhina et al. 2016, Plant Phys (november). Despite these differences, the thylakoid stacks of these two organisms appear to share several properties (as discussed above).

The Reviewer is right. In the first version of the manuscript, we did not compare diatoms with *Chlamydomonas*, since this type of comparison (green vs red algae) is sometimes criticised by the community. However, following her/his suggestions, we have decide to draw a parallel between our results and previous data in *Chlamydomonas*. Data are discussed on pages 6 and 10 of the revised manuscript, where relevant references are quoted (new references 14, 29-32). We

would like to thank the Reviewer for this useful suggestion, since we believe it will stimulate both discussion and further work in the area.

e. The measurement of thylakoid lumen width should also be compared to *Chlamydomonas*. See the detailed explanation in point 2 directly below.

Answer to points 1e to 2d. We agree with the Reviewer that this is a rather complex question that is not adequately answered. This point has also been criticised by other Reviewers. It is true that our data have been obtained in conditions that differ from the ones used to study *Arabidopsis* and *Chlamydomonas*. Therefore, it is not possible to directly compare these findings. As we might have over-interpreted our data in the previous discussion, and because we do not presently have suitable new data, we have decided to remove this part in the revised version of the manuscript.

2) : Line 135: *“To explain the different equilibration time of diatoms and plants, we reexamined the EM pictures of samples prepared with the Tokuyasu protocol. By preserving the membrane structures, this technique allowed additional features of the *P. tricornutum* thylakoids to be observed. First, we found that the size of the *P. tricornutum* lumen (7.1 ± 1.5 nm, average of 100 estimates from 67 different preparations) was larger than that of plants (4.5 nm), consistent with the expected loose membrane stacking in diatoms. This size is double the size of the soluble electron carrier *cyt c6* ($\sim 33 \times 23$ Å), while in plants the lumen size is equivalent to that of the soluble redox carrier *plastocyanin*. The larger lumen size in *P. tricornutum* should facilitate the diffusion of the soluble carriers.”*

These conclusions are misleading and require changes:

a. In higher plants, thylakoid lumen width is dynamic and swells from 4.5 nm in dark-adapted grana (Kirchhoff et al. 2011, PNAS; Daum et al. 2010, Plant Cell) to 9 nm in plants that have been adapted to the light (Kirchhoff et al. 2011). In other words, when the light reactions of photosynthesis are active, the plant thylakoid lumen is 9 nm. Thus, it is not true that diatoms have more luminal space for the diffusion of soluble carriers.

b. When grown in the light, *Chlamydomonas* also has a thylakoid lumen width of 9nm (Engel et al. 2015). It is currently untested whether this width shrinks in the dark, like higher plants. Regardless, *Chlamydomonas* also has sufficient luminal space for the diffusion of soluble carriers.

c. Later in the manuscript, the Kirchhoff et al. observation is finally cited. However, it is done in a confusing and conflicting way. Line 178: *“Finally, a size of the lumen larger than the size of the soluble electron carrier plus the presence of interlinking thylakoids should facilitate diffusion of *cyt c6*, thus allowing fast redox equilibration between the photosynthetic complexes in the diatom, unlike*

plants. Consistent with this idea, lumen widening has been observed in plants exposed to high light to facilitate electron flow and prevent photodamage in plants”

These two sentences directly conflict with each other. The authors say that the diatom lumen is wider, thus enabling rapid diffusion and equilibrium of soluble carriers, unlike in plants. And then they say that the expansion of the plant thylakoid lumen in light is consistent with this idea. However, it is not. As the plant thylakoid lumen is at least as wide as the diatom lumen during light-driven photosynthesis (or maybe even wider: 9 nm vs. 7 nm), a difference in luminal width cannot explain the difference in redox equilibrium rates between diatoms and plants. In contrast, it does seem plausible that the increased interconnectivity provided by “linking” thylakoids improves diffusion in diatoms.

d. Given the published luminal widths from Kirchhoff et al., Daum et al., and Engel et al. (each of which should be clearly cited and described in the manuscript), how do the authors interpret their measurements in diatoms? Importantly, what were the photosynthetic conditions of the cells when they were fixed in aldehyde for electron microscopy? The 7 nm diatom measurement falls in between the 9 nm high-light and 4.5 nm dark luminal widths reported for plants and *Chlamydomonas*. Were the diatoms grown in medium or low light (please contextualize 20 $\mu\text{mol photon m}^{-2}\text{s}^{-1}$ with 12 hr light/dark cycle), and thus were the cells performing an intermediate level of photosynthesis? This might make sense with a 7 nm width, assuming thylakoid dynamics are consistent between plants, *Chlamydomonas*, and diatoms. Additional dark and high-light measurements of the diatom thylakoids would be very informative. However, if those experiments cannot be accomplished for this publication, the existing data should be put in the proper context.

3) In Figs. 4 and 5, real images from the FIB/SEM 3D volumes must accompany the segmentations that are shown. While segmentations are helpful for interpreting 3D morphology, they are only an interpretation. It is absolutely essential that features in the actual micrographs are displayed for the readers. 3D volumes can be rotated and sliced in software such as IMOD’s 3dmod slicer window to show precisely the features of interest in the desired orientation.

Following the Reviewer’s advice, we have modified both Fig. 4 and Fig. 5. In the revised versions of these figures real images (EM micrographs) are presented along with the 3D reconstruction after segmentation. We have also modified the figure legends, in order to provide a more clear explanation of the colour code employed in these reconstructions.

a. In Fig. 4, there should be a new first panel that shows a longitudinal end-to-end overview slice of the cell, rotated from the real FIB/SEM data so that the whole cell can be seen in one image. This image should be in the same orientation of the current panel A, which will then become panel B.

This panel is now displayed in the new Fig. 4. The figure legend has been modified accordingly.

b. In Fig. 4, a FIB/SEM close-up image of the “evagination” contact between the chloroplast and nucleus is needed to accompany panel D. Only by seeing the real data can we judge the nature of the membrane contact. This contact is also not very evident in the view of the segmentation that is shown. A better view should be displayed, perhaps with an arrow pointing to the contact.

As suggested, an image showing the membrane contacts between the nucleus and the chloroplast is now included in Fig. 4c. The figure legend has been modified accordingly.

c. In Fig. 5, two FIB/SEM close-up images are needed to accompany the segmentations of “linking thylakoids” in panels B and C. The black and red arrows pointing to thylakoids and plastoglobules in panel C should also be indicated on the FIB/SEM micrographs. This is a potentially novel structure, so it is very important that the readers are shown what it looks like in the actual EM data.

We have included SEM micrographs and a new image showing the linking thylakoids in the Fig. 5 of the revised manuscript. The figure legend has been modified accordingly. However, despite new data (see e.g. new supplementary movie 2 with a 2 nm voxel), our resolution is still imperfect for visualising structures of this size, as highlighted by Reviewer, 4 below.

Minor points that should also be addressed:

1) The authors use the names PS1 and PS2, but I believe the standard nomenclature is PSI and PSII. This should be changed throughout the manuscript.

This change has been done in the revised manuscript.

2) Line 94: “*we immunolocalized the two PSs in cells prepared using the Tokuyasu protocol, an optimal method to preserve membrane structures for electron microscopy (EM) imaging*”

As stated in the methods, the Tokuyasu protocol involves standard aldehyde fixation of the biological material, which is known to cause deformations of membranes. This is certainly not an “optimal” method to preserve membranes. While the optimal approach would be FIB-thinning of vitreous frozen-hydrated cells, the best option for conventional resin-embedded EM would be high pressure freezing followed by freeze-substitution. I do not believe that the aldehyde fixation used by the authors invalidates any of their results, but the wording should be changed to: “we immunolocalized the two PSs in cells prepared for electron microscopy (EM) using the Tokuyasu protocol, a cryo-sectioning method that improves the preservation of membranes structures”.

The sentence has been modified in the revised manuscript.

3) The biochemical prep in Fig. S4 is one of the major lines of evidence used by the authors to conclude that PSI and PSII are partitioned to specific thylakoid compartments. Thus, this data should not be in the supplement, but rather should be added to Fig. 2 to accompany the immuno-EM localization.

Western blots are now included in the new Fig. 2g. Please note that this panel displays new data, also addressing the features of cytochrome *b₆f*. The figure legend has been modified accordingly.

4) The labels “d” and “f” in Fig. 1, “b” and “d” in Fig. 3, and “c” in Fig. 5 should be moved to the upper left of the corresponding panels. This is the standard label position for scientific figures.

These mistakes have been corrected in the revised version of the manuscript.

5) In Fig. 1A, the label “common antenna” should be changed to “shared antenna”. The word “common” has a common secondary meaning, which I just used in this sentence.

The word “common” has been changed in the revised version of the manuscript.

6) When Fig. 2 is printed, the dark blue color denoting the envelope is very similar to the purple color showing the “core” membranes. The dark blue should be changed to light blue or magenta.

The colours have been changed in the revised version of Fig. 2, and the figure legend and text modified accordingly.

7) In Fig. 3, the overlapping shape symbols make the plots in panels B, C, and D quite hard to read. A less cluttered plotting method with colored or patterned lines might be preferable.

Colors have been introduced in the new version of Fig. 3.

8) The phrase “lateral heterogeneity” should be used at least once in the manuscript to describe the segregated localization of PSI and PSII to different thylakoid regions.

This is a very good suggestion. We have introduced the concept of lateral heterogeneity at the beginning of the introduction (page 3) in the revised version of the manuscript.

9) The English prose in this manuscript would benefit from a good polishing.

I list some edits below, but this list is not exhaustive:

a. Remove “i.e.” from lines 57, 59, 116, 128, 130, and 173. Each of these sentences makes sense and is easier to read without “id est” getting in the way. :

b. Throughout the manuscript, “via” should not be italicized. Although from Latin origin, it is used in standard English.

- c. Line 58: “Primary plastids comprise differentiated thylakoid domains” Change to: “Primary plastids contain differentiated thylakoid domains”
- d. Line 73: “a sophisticated thylakoid membrane network that orchestrates photosynthetic light absorption and utilisation via subtle subthylakoid segregation of the PSs.” Change to: “a sophisticated thylakoid membrane network that orchestrates photosynthetic light absorption and utilisation via segregation of the PSs to specific thylakoid subdomains.”
- e. Line 78: “thereby favouring energy spillover via physical contacts.” Change to: “thereby favouring energy spillover via physical contacts between the complexes.”
- f. Line 82: “This phenomenon has been documented in PS2-poisoned red algae and interpreted as a signature of spillover in these algae, considered the ancestors of secondary plastids.” Change to: “In red algae, considered the ancestors of secondary plastids, this phenomenon was observed upon poisoning of PSII and was interpreted as a signature of spillover.”
- g. Line 91: “Thus, either (i) physical barriers-created by lipid/biochemical surroundings-prevent energy exchange” Change to: “Thus, either (i) lipid or biochemical barriers prevent energy exchange”
- h. Line 102: “In plant thylakoids (see e.g.12)” Change to: “In plant thylakoids¹²”
- i. Line 116: “an ‘equilibration plot’ (Fig. 3), i.e., the relationship” Change to: “an ‘equilibration plot’ (Fig. 3), which shows the relationship”
- j. Line 149: “focused ion-beam” Change to: “focused ion beam”
- k. Line 150: “We identified the organelles and their interactions” Change to: “By segmenting the 3D volume, we identified the organelles and their contacts”
- l. Line 158: “A focus on the 3D structure of the photosynthetic membranes (Fig. 5a-c) confirmed the presence of parallel layers of thylakoids in the plastid but also revealed the presence of thylakoids (Fig. 5b,c, green) connecting the different thylakoid layers” Change to: “The 3D structure of the photosynthetic membranes (Fig. 5a-c) confirmed the presence of parallel layers of stacked thylakoids, but also revealed the presence of single thylakoids (Fig. 5b,c, green) connecting the different thylakoid stacks”
- m. Line 163: “Our 3D reconstruction of the *P. tricornutum* plastid with FIB-SEM discloses” Change to: “Our 3D FIB-SEM reconstruction of the *P. tricornutum* plastid reveals”
- n. Line 167: “Subtle compartmentalisation” Change to: “compartmentalisation”. There is nothing in the data to indicate that it is subtle.

o. Line 173: “state transitions... does not exist in diatoms” Change to: “state transitions... do not exist in diatoms” And is this a certainty? It may be better to say: “state transitions... are believed to not exist in diatoms”

All these suggestions have been considered and the corresponding changes made in the text. Moreover, the whole text has been revised for clarity and typos. Many thanks to Reviewer 3 for her/his help.

Reviewer #4 (Remarks to the Author):

The manuscript by Flori et al. deals with one of the longstanding questions in research on eukaryotic algae belonging to the ecologically extremely important group of secondary endosymbionts, the diatoms. In contrast to higher plants thylakoids are not separated in grana and stroma thylakoids, but organised in lamellae of mostly three thylakoids each spanning the whole plastid. This poses the question of energetic separation of photosystem II and I and the feasibility of electron transport in such an organisation. Thus, the manuscript deals with a very important question in photosynthetic research on this important group of organisms. The manuscript is tackling three main research questions, i) is there spill-over (direct energy transfer) from PSII to PSI, ii) are the photosystems equally distributed over the six membranes of the lamellae, and iii) are there connections between single thylakoids in a/between lamellae enabling electron transfer between them.

First I would like to discuss the novelty of the findings:

i) **Spill-over was not explicitly addressed in diatoms before.** However, there are several studies on excitation energy transfer in vivo, and neither found strong indications for a huge amount of spill-over. Miloslavina et al. *Biochim. Biophys. Acta* 1787(10), 1189-1197 (2009) discuss only one of their minor components in this context, as do Yokono et al. (Ref 10 in the manuscript), whereas Chukhutsina et al. *Biochim. Biophys. Acta* 1827(1), 10-18 (2013) did experiments with and without DCMU like done here and the lifetime of PSI did not change. Although this was not explicitly discussed as a lack of spill-over there is no other explanation for it. However, nobody measured spill-over in diatoms on the basis of electron transport before as done here.

The Chukhutsina et al. article is now cited in the manuscript (new reference 12). However, as recognised by the Reviewer, this article does not focus on spillover. The only publication addressing the occurrence of spillover in diatoms is that of Yokono and coworkers. There, the authors concluded that spillover occurs in *P. tricornutum*, which is challenged by our results. Therefore, we do not agree with the Reviewer when she/he concludes that our data on spillover simply constitutes a confirmation or obvious interpretation of previously published data (see also below). This is new information, which provides a sound explanation for previously published data (i.e. the fluorescence decay analysis of Chukhutsina et al.). Finally, at variance with other articles investigating spillover in algae (e.g. the recent work in *Symbiodinium* now quoted as reference 10), our work provides additional information than spectroscopy. This is required to understand the structural bases for the presence of absence of spillover. This is also entirely new.

ii) The results of Ref 7 are not cited entirely: this study was the first to detect an uneven distribution of PSI with a preference for the outer membranes of a lamellae using immunolabelling in electron

microscopy, but for whatever reason authors do not refer to this major result of the paper. PSII was never tested, and although the hypothesis of PSII being located in the inner membranes is not new, nobody has ever proven it experimentally before.

We thank the Reviewer for noticing this mistake. The results published by Pyszniak and Gibbs (1992) are now properly discussed in the revised manuscript (Ref 6).

iii) so far only transmission electron microscopy data obtained after thin sectioning exist about the thylakoid structure of diatoms. Thus, every higher resolution data or 3D reconstruction is of great value. However, my main concern is about the interpretation of the FIB-SEM data on thylakoid 3D structure. According to Material and Methods the data were acquired with a pixel size of 4x4 nm, and removal of about 4 nm for each slice in z-direction, ending up in roughly 4x4x4 nm voxels. A membrane has a thickness of 4-5 nm. By imaging using the described voxel size, single membranes are not to be distinguished. Even for a whole thylakoid (about 10 nm for the two membranes, plus around 7 nm for the lumen) the resolution is beyond its limits with only four voxels in z-direction. As a consequence, Fig. 5b and c show noisy surfaces of unknown attribution, but shed no light on the interesting question of thylakoid 3D structure in diatoms. Also for the really nice TEM images shown here, slices were 80 nm, so no distinction between overlays and crossings can be made (Fig. S5) due to the sample thickness. For both methods cells were fixed with glutaraldehyde, which was often discussed as critical for measuring exact membrane distances. Also the finding that there are not always three thylakoids per lamellae, but sometimes 'suddenly' four or only two is old (Ref 5 and 7). Some further major concerns can be found below.

The Reviewer is right when she/he says that that our reconstruction of the 3D structure of the chloroplast is a low resolution one. This is an intrinsic limitation of the FIB-SEM approach. Supplementary movie 2 shows a new stack acquired with a voxel size of 2 nm, that with is a SEM pixel size of 2 nm and FIB slices of 2 nm. Although this resolution is similar to the theoretical maximum resolution of our SEM at 1.5 kV (as used in our work), the results are not significantly better. Two reasons possibly explain this finding:

i. when the resolution is enhanced, the electron dose per surface unit during the acquisition is also increased (approximatively by a factor of 4). This leads to higher electron beam fluctuations during the acquisition and/or thermal damages to the sample.

ii. simulations suggests that in our conditions (1.5 kV, with a EsB grid at 1 kV) the recorded backscatter signals emerge primarily from an escape depth or around 5 nm (see e.g. Narayan and Subramaniam (2015), quoted as reference 39), but certainly

higher than 2 nm, Jouneau , unpublished). This of course also limits the final resolution of the method.

It seems therefore that, in our case, a pixel size of 4 nm likely represents the best compromise.

Consequently, the “membranes “presented in Fig. 5 (violet surfaces) do not represent single thylakoids but layers of thylakoids. Nonetheless, the resolution of these images is still sufficient to highlighting the presence of connections between these layers (indicated by yellow circles in the new Fig. 5). These connections can easily be distinguished from plastoglobules, which have a higher density in our images. The thylakoid connections can also be visualised on single SEM micrographs, which are now included in the revised figure of Fig. 5 as advised by Reviewer 3. Our 3D reconstruction also reveals a peculiar 3D arrangement of the photosynthetic membranes around the chloroplast pyrenoid, again confirming that thylakoids in diatoms are structured, and not simply made of three layers of loosely stacked membranes, as often stated in the literature. It seems that the thylakoid structure in diatoms (and possibly in other organisms arising from secondary endosymbiosis) is close to that observed in chlorophytes (as suggested by Reviewer 3).

Overall, several important and novel conclusions can be drawn based on these 3D structures: i. thylakoids in diatoms are highly structured, ii. their structure closely resembles the one observed in plants and green algae, suggesting a functional convergence towards a system capable of optimising photosynthesis and reducing photon waste. iii. this structural arrangement is fully compatible with the biochemical data (Fig. 2) and also with the functional assessment of the efficiency of photosynthetic electron flow (Fig. 3). All these points were already discussed in the previous version of the manuscript. However, in response to the concerns of Reviewer 4 we have improved the presentation and revised our discussion about the structural data. In the revised version, these data are more cautiously interpreted than in the previous version of the manuscript, and the conclusions that can be derived from these data are better presented (page 9).

Thus, novelty is limited and authors should be more careful in the description of whether data they present are confirming former publications (no spill-over, PSI localisation, nucleus-chloroplast interaction) or new (PSII localisation). Most importantly they should limit their interpretation of electron microscopy data to the resolution obtained or better, acquire higher resolution data.

Once again, it is true that the PSI localisation was previously assessed in diatoms. However, knowing the localisation of a single photosystem does not allow any hypothesis on the energetic interactions between PSI and PSII to be made, which is the basis to interpret the lack of spillover. This is the reason why we tested the localisation of the two PSs in our work, and the

robustness of our data has been substantially improved in this revised version. Based on the biochemical and immunolabelling results we can offer a sound explanation for our functional analysis. This is new. Even in the recently published study about spillover in a related organisms *Symbiodinium* (reference 10 in the revised manuscript), the authors can only speculate about possible rearrangements of PSI and PSII to interpret their data, because no biochemical/structural relationship is provided to address this hypothesis. Overall, the main goal of our article is not to perform a structural study of the diatom chloroplast, but to propose a realistic scenario of diatom photosynthesis based on complementary pieces of evidences (biochemistry, spectroscopy, immunolabelling, 3D reconstruction). Because of the nature of some of the Reviewer's comments we feel that this goal was not conveyed adequately in the initial version of the manuscript. Thus, we have revised several parts to better highlight the main goal and achievements of this work (see also the answers to the other Reviewers).

Further major points

1) The nice spectroscopic analysis of P700 and cyt oxidation shows no deviation between curves measured with or without DCMU (+HA) in Fig. 1 of the main text. However, in the supplementary file data obtained using different light intensities are shown. In Fig. S2 it is obvious that the deviation between data obtained with and without DCMU+HA is strongly light intensity dependent, with a maximum of about 20% at the highest intensity. This needs explanation and discussion, since 20% is far from negligible or 'limited' (line 90). In addition there is a contradiction between Fig 1 and Fig S2f: according to the legends the same light intensity was used but graphs differ tremendously, which light intensity was used in Fig. 1?

The Reviewer is right; there was a (partial) contradiction between the different experiments. For this reason, measurements of the light dependency of PSI turnover have been repeated, and the new results are presented in Fig. 1 and Fig. S2.

In order to improve clarity, results relative to the highest light intensity are presented in Fig. 1, while data relative to the other light intensities are shown in the Fig. S2. The corresponding figure legends have been modified accordingly. Please note that while repeating these experiments we realised that the highest light intensity was lower ($800 \mu\text{mol photons m}^{-2} \text{s}^{-1}$) than the value indicated in the previous version of the manuscript ($1100 \mu\text{mol photons m}^{-2} \text{s}^{-1}$). No changes were found in the case of the other intensities. To acknowledge this fact, the legend of Fig. 1 has been modified.

2) Due to the spectroscopic overlap it is only possible to probe the redox state of cyt c and cyt f simultaneously, as rightly stated in Material and Methods. However, in the Results authors always use

cyt c, which is misleading. Please use either cyt or [cyt c + cyt f]. This is especially important in the sections about diffusion barriers, since the calculations cannot discriminate between cytf/cyt c and cyt c/PSI electron transfer. In addition, please mention the light intensity used in the legend of Fig. 3

Following her/his advice, we now consistently use “cyt” in the revised version. The light intensities used for Fig. 3 is now specified: high light (800 $\mu\text{mol photons m}^{-2} \text{ s}^{-1}$) was used for redox spectroscopy, to ensure that electron flow was maximum. Low light (18 $\mu\text{mol photons m}^{-2} \text{ s}^{-1}$) was used in the case of fluorescence measurements. At this intensity, fluorescence does not reach the maximum value (F_m) in the absence of DCMU, allowing to properly assess the inhibition of PSII by the inhibitor. The legend of Fig. 3 has been modified accordingly.

3) Estimations of the lumenal space or the distance between thylakoids from EM data are always hampered and were often criticised in cases where fixatives were used like in this study. In Ref 7 already 6 nm were reported. SANS studies like those of Nagy et al. Biochem. J. 436(2), 225-230 (2011) revealed 17 nm for a whole thylakoid in vivo. If taking the usual estimate for membrane thickness (4-5 nm) this would leave maximal 9 nm for both the space between thylakoids and the lumen. Those values should at least be discussed.

As already mentioned in the answer to Reviewer 3 (answer to points 1e to 2d), we also agree with Reviewer 4 that this part of the discussion was not fully supported by the data. Therefore, we have decided to remove it in this revised version of the manuscript, as it is not central to the arguments.

4) FIB-SEM, line 153-157: The contact between nucleus and chloroplast was to be expected, since the outer membrane of the chloroplast envelope in heteroconts is in connection with the nuclear ER due to their evolutionary history. These features are quite different to the cited plant stromules. For Fig. 5 the scale bar is missing and it would be nice to know what and why certain areas were shaded green and purple in b and c.

The Reviewer is right, the nucleus/chloroplast contact in diatoms has different features than the ones of the stromules found in plants. Nonetheless, from a functional point of view, this interaction could facilitate exchanges between the two compartments as recently proposed in plants (ref. 21) for the stromules. We have therefore modified this sentence (page 8) simply to acknowledge this fact.

Scale bars are present in the new Fig. 5, which has been completely modified following the suggestions of Reviewer 3, and the colour code is now explained in the figure legend.

5) Material and Methods suffers from lengthy descriptions of e.g. standard statistical methods (PCA), but many important details are missing: When was DCMU added in the different experiments? It

seems that for cytochromes extinction coefficients were used, whereas PSI oxidation is plotted on a relative scale, please explain/mention.

The text describing the PCA analysis has been modified in the revised manuscript, to explain the new type of analysis, which considers 5 antibodies instead of the 2 previously employed. We have also checked that significant information was not missing in the text. In particular, we now state that DCMU was added immediately before measurement as indicated in the legend of Fig. 1 of the revised manuscript and in Fig. 3. Successful inhibition was tested in both cases by a direct measurement of the fluorescence rise kinetics. Because of the short time required to perform these experiments (a few minutes for every sample), we can rule out additional consequences of DCMU on the photosynthetic apparatus and/or on the thylakoid structure, which could take place on a longer time scale.

Finally, no extinction coefficients are needed to evaluate the PSI/cyt stoichiometry in our case, since this parameter is directly evaluated based on the ratio between the amount of oxidised cyt and PSI. As now explained in the text (page 13) and in the legend of Fig. S1 we compare the amount of cyt that is oxidised by a single PSI turnover (laser flash) with the amount that can be oxidised in continuous light, in a sample where electron flow is prevented by DCMU. Based on this ratio (1/3) we can calculate the cyt/PSI stoichiometry.

The rate of generation of electrons by PSII (Results, lines 127 – 134 and Fig. 3) was probably done as in Ref 3, i.e. using the dark relaxation. This should be mentioned.

The approach employed to evaluate the photosynthetic rate is based on the relaxation kinetics; this is mentioned in the methods (page 12) and in the legend of Fig. S1.

Intactness of chloroplasts is given as 70%. The method used tests the intactness of the envelope (penetration of ferricyanide). For an estimate of overall intactness the absolute O₂ evolution rates have to be given in comparison to the rates seen in cells.

The Reviewer is right, there is no comparison between the rates measured in chloroplast and in vivo. However, the ferricyanide method is largely accepted by the community as a test for intactness. In any case, the purpose of this work is not to characterise intact chloroplasts from diatoms, but only to perform the biochemical analysis on samples where the thylakoid structure is preserved. In our view, intact chloroplasts represent a good starting material to perform these experiments.

Line 280/281: are these the concentrations of the detergent solutions or the final concentrations? If the former is true, how much was added?

As explained in the Method section of the revised manuscript, the final detergent concentrations are given.

Which method was employed to measure protein concentration? Which gels were used for the final analysis (7% or 13%)?

Protein concentration was measured using a commercial Bradford protein assay kit. However, the new data presented in the Fig. 2 have been obtained using a modified protocol to improve the results. We used a 4-20% SDS PAGE and we loaded samples on a chlorophyll basis. These changes are now detailed in the Method section.

Which secondary antibody was used for the gold-labelling, and, most importantly: what was the size of the gold label?

We apologise for not having provided this information in the first version of the manuscript. Two different secondary antibodies were employed with the 5 antibodies against PSI, PSI and the *cyt b₆f* complex. The size of the gold labels is now provided in the methods section. In the new Fig. S4 we also provide evidence that none of the secondary antibody cross reacts with the thylakoids, as requested by Reviewer 1.

Authors did a sophisticated statistical analysis, but how were the images analysed in order to acquire the basic data set? Antibody labelling is often seen in areas belonging to the lumen of the outmost thylakoid (e.g. Fig. 2c, lower right corner). Were these labels neglected, counted to outer membranes or core membranes?

The statistical analysis provided in the revised version of the manuscript is based on manual counting of around 6000 gold particles. Samples were analysed before knowing the nature of the antibody employed. First, the total number of labels was assessed and then particles were attributed to various compartments. If gold particles were uncertainly located (including labelling in the luminal space, as mentioned by the Reviewer), or found at a place where the quality of the picture was not sufficient to appreciate the thylakoid ultrastructure, they were not for considered for the statistical treatment. The number of discarded beads is around one third of the total (3995 beads out of 11932). In most cases, the manual analysis was repeated by another person (with the same protocol). Probably because of the very high number of micrographs analysed, the two analyses converged to very similar results. The entire procedure is now explained in the method section (page 16).

Fig S3: are the scale bars correct? They imply that a factor of 2.5 is in between both pictures – the difference in size of e.g. thylakoid lamellae looks less

The Reviewer is entirely right. We apologise for this mistake, which has been corrected in the revised version of Fig. S3.

Minor points

Line 67/68: deviations from the 3 thylakoid per lamellae scheme were already reported by Refs 5 and 7. Thus the sentence would be more informative if rephrased to ‘loose stack of mostly three thylakoids, in some cases two or four, with a few...’ and both references cited, since also anastomoses were detected by Ref 7

The two references are properly quoted in the revised version of the manuscript.

Line 78: Ref 7 does not show a random distribution of PSI, on the contrary (see above). Only FCPs were randomly distributed

Line 97: the preferential localisation of PSI in the outer membranes requires a reference to Ref 7 (‘as shown before’)

The reference Pyszniak and Gibbs, (ref 6 in the revised manuscript) is now appropriately cited.

Line 97/98: in line 97 is should read Fig 2b, c, whereas in line 98 Fig 2a, c is correct

This has been corrected in the new version of the manuscript.

Line 146: anastomoses are not ‘unexpected’, see for example Ref 5 and 7

The term “unexpected” has been removed.

Line 166: The paper cited (Ref 18) has nothing to do with diatoms and their thylakoid structure

In the previous version of the manuscript, reference 18 was cited on purpose, to highlight the fact that even in publications referring to general aspects of algal physiology, the notion of a simplified thylakoid structure in chloroplasts arising from secondary endosymbiosis is mentioned. But we agree that this review is not the most pertinent one. We now cite reference 22 instead, where these aspects are discussed in more details.

Line 169: nobody measured lipid distribution, so the observations are compatible ‘with the hypothesis that the core membranes are enriched in lipids...’

The Reviewer is right, the sentence has been modified.

Line 178: there is indeed a group of organisms with similar thylakoid structure (dinoflagellates) where recently a huge spill-over was reported under high light conditions (Slavov et al Biochim. Biophys. Acta 1857(6), 840-847 (2016))

Many thanks for this suggestion. The reference has been cited (new reference 10), and the conclusions discussed in the revised version of the manuscript. We also mention additional results (Bina et al 2016, reference 28), showing that prolonged exposure to far-red light induces accumulation of PSI in specific domains in diatoms, because these data are also relevant for our model.

Line 211: wrong format for multiplication in formula

This mistake has been corrected in the revised version of the manuscript.

Line 223: ‘...reduction rate of this electron donor pool’ would be more clear

The sentence has been modified in the revised version of the manuscript.

Line 230: last word ‘not’

This has been changed in the revised version of the manuscript.

Line 257: should be in bold

This mistake has been corrected in the revised version of the manuscript.

Line 299: what does (10) stand for?

This mistake has been corrected in the revised version of the manuscript (reference 13).

Line 339: was

This has been changed in the revised version of the manuscript.

Line 373: which appendix?

The appendix was missing in the first version of the manuscript. We have added it to the revised text, at the beginning of the supplementary file.

Line 450: light (not fight).

This mistake has been corrected in the revised version of the manuscript.

Refs 21 and 22 seemed to be swapped

This mistake has been corrected in the revised version of the manuscript.

Reviewers' comments:

Reviewer #1 (Remarks to the Author):

The essential message is that PSI and PSII are spatially separated, while b6f is more homogeneously distributed. The electron transfer data suggest that while there is some disequilibrium between cyt and PSI redox state this is smaller than in plants. The authors suggest that the presence of the membrane bridges in diatoms promotes fast equilibration of cyt c6 between the membrane layers.

I still think this section of the manuscript requires further explanation. If the b6f is mainly in the PSI regions how do the membrane bridges help? are they only helpful to the c6 diffusing from the b6f in the PSII regions? Do the bridges also facilitate PQ diffusion from PSII to b6f in the PSI regions? These points are not clear. It is also unclear why with increased electron flow in Fig. 4C that the disequilibria should get worse. I would anticipate that increased electron flow would lead to expansion of the lumen so facilitating redox equilibration, as shown by Kirchoff in plants. These results seem to suggest the opposite?

Reviewer #2 (Remarks to the Author):

The authors have appropriately addressed all points I raised. Therefore, I would recommend publication of this interesting work.

Reviewer #3 (Remarks to the Author):

The revised manuscript satisfactorily addresses the points raised during the previous round of review, and I believe it is suitable for publication. The journal has my permission to release my name along with my reviewer comments upon publication (Benjamin Engel, reviewer #3).

Reviewer #4 (Remarks to the Author):

The revised manuscript by Flori et al. has extremely improved by more careful interpretation of the data, the additional experiments that were included and the more detailed citations.

Thus, in my point of view all former concerns were dealt with and there is only one minor point: in the answers to one of the referees the number of particles used for the statistical analysis of the localisation of PSI and PSII is given – it would be nice if readers would know this number as well, i.e. the number of particles finally used for the PCA.

Answers (in red) to Reviewers' comments (in black):

Reviewer #1 (Remarks to the Author):

The essential message is that PSI and PSII are spatially separated, while b₆f is more homogeneously distributed. The electron transfer data suggest that while there is some disequilibrium between cyt and PSI redox state this is smaller than in plants. The authors suggest that the presence of the membrane bridges in diatoms promotes fast equilibration of cyt c₆ between the membrane layers.

I still think this section of the manuscript requires further explanation. If the b₆f is mainly in the PSI regions how do the membrane bridges help? are they only helpful to the c₆ diffusing from the b₆f in the PSII regions? Do the bridges also facilitate PQ diffusion from PSII to b₆f in the PSI regions? These points are not clear.

The reviewer is right: the more homogeneous distribution of the cytochrome *b₆f* complex, which is present in both the core and the peripheral membranes (although preferentially in the latter) suggests that the membrane bridges should play a dual role. They could allow short distance diffusion of the cytochrome c₆ between the *b₆f* complexes in the core membranes and PSI, but also short distance diffusion of the plastoquinones between the PSII in the core membranes and the *b₆f* complexes in the peripheral membranes. This idea is now explicitly mentioned in a new paragraph (page 10 in the merged pdf, lines 207-213). Note however that while our data show unambiguously that the equilibration between cytochromes and PSI is very efficient, we do not have experimental evidence for a very fast diffusion of PQ between PSII and the *b₆f* complex.

It is also unclear why with increased electron flow in Fig. 4C that the disequilibria should get worse. I would anticipate that increased electron flow would lead to expansion of the lumen so facilitating redox equilibration, as shown by Kirchoff in plants. These results seem to suggest the opposite?

In order to see a kinetic limitation of electron flow by diffusion domains -if any- one needs to work under condition where the electron flow rate is faster than the equilibration time between cytochromes and PSI. This situation is in principle not expected under light-limited photosynthesis, where the rate of electron flow is set by light harvesting by PSII and PSI, and not by diffusion between cytochromes and PSI. As suggested by the reviewer, limitation of electron flow by restricted diffusion could nonetheless be seen in these conditions, provided that the (small) size of the lumen approaches that of cytochromes c₆, leading to restricted

diffusion. However, preliminary experiments performed at low light suggest that this is not the case. This might suggest that the regulation of the luminal size in diatoms is somehow different than in plants. However (as already acknowledged in the case of our previous answer to the reviewers), we have not enough elements at the moment to properly answer this question.

On the other hand, we expected to see (and actually observed in Fig. 3) limitation of photosynthesis by restricted diffusion in light-saturating conditions. Of course, we agree that under light saturated conditions, strategies are developed to alleviate limitation of photosynthesis, including e.g. the swelling of the lumen reported by Kirchhoff and coworkers in plants. But these responses are integrated by our estimation of electron flow in vivo, which provides a robust evidence for the existence of diffusion domains with high equilibration rates under truly physiological conditions.

We agree with the reviewers that this part was not clear enough in the revised version of the manuscript. Therefore we have added a new sentence in the revised manuscript (page 14 in the merged pdf, lines 293-296) to provide a rationale for the experiments of Fig. 3.

Reviewer #2 (Remarks to the Author):

The authors have appropriately addressed all points I raised. Therefore, I would recommend publication of this interesting work.

Many thanks to Reviewer 2

Reviewer #3 (Remarks to the Author):

the revised manuscript satisfactorily addresses the points raised during the previous round of review, and I believe it is suitable for publication. The journal has my permission to release my name along with my reviewer comments upon publication (Benjamin Engel, reviewer #3).

Many thanks to Reviewer 3

Reviewer #4 (Remarks to the Author):

The revised manuscript by Flori et al. has extremely improved by more careful interpretation of the data, the additional experiments that were included and the more detailed citations. Thus, in my point of view all former concerns were dealt with and there is only one minor point: in the answers to one of the referees the number of particles used for the statistical analysis of the localisation of PSI and PSII is given – it would be nice if readers would know this number as well, i.e. the number of particles finally used for the PCA.

The numbers of particles employed for the PCA analysis is now explicitly mentioned in the methods section (page 17 in the merged pdf, lines 361-363)

REVIEWERS' COMMENTS:

Reviewer #1 (Remarks to the Author):

I am satisfied with the answers given.